**Data Availability Statement:** Satellite data cannot be shared publicly because of its sensitivity under high resolution. However, they are available from

# Ecological factors associated with persistent circulation of multiple highly pathogenic avian influenza viruses among poultry farms in Taiwan during 2015-17

**Wei-Shan Liang[1], Yu-Chen He[1], Hong-Dar Wu[2], Yao-Tsun Li[3], Tai-Hwa Shih[4], Gour-Shenq Kao[4], Horng-Yuh Guo[5], Day-Yu Chao📧[1] ***

**1** Graduate Institute of Microbiology and Public Health, College of Veterinary Medicine, National Chung-Hsing University, Taichung, Taiwan, **2** Institute of statistics, National Chung Hsing University, Taichung, Taiwan, **3** Program in Emerging Infectious Diseases, Duke-NUS Medical School, Singapore, Singapore, **4** Bureau of Animal and Plant Health Inspection and Quarantine (BAPHIQ), Taipei, Taiwan, **5** Division of Agricultural Chemistry, Taiwan Agriculture Research Institute (TARI), Council of Agriculture, Taichung, Taiwan

* dychao@nchu.edu.tw

## Abstract

Emergence and intercontinental spread of highly pathogenic avian influenza A (HPAI) H5Nx virus clade 2.3.4.4 has resulted in substantial economic losses to the poultry industry in Asia, Europe, and North America. The long-distance migratory birds have been suggested to play a major role in the global spread of avian influenza viruses during this wave of pan-zootic outbreaks since 2013. Poultry farm epidemics caused by multiple introduction of different HPAI novel subtypes of clade 2.3.4.4 viruses also occurred in Taiwan between 2015 and 2017. The mandatory and active surveillance detected H5N3 and H5N6 circulation in 2015 and 2017, respectively, while H5N2 and H5N8 were persistently identified in poultry farms since their first arrival in 2015. This study intended to assess the importance of various ecological factors contributed to the persistence of HPAI during three consecutive years. We used satellite technology to identify the location of waterfowl flocks. Four risk factors consistently showed strong association with the spatial clustering of H5N2 and H5N8 circulations during 2015 and 2017, including high poultry farm density (aOR:17.46, 95%CI: 5.91–74.86 and 8.23, 95% CI: 2.12–54.86 in 2015 and 2017, respectively), poultry heterogeneity index (aOR of 12.28, 95%CI: 5.02–31.14 and 2.79, 95%CI: 1.00–7.69, in 2015 and 2017, respectively), non-registered waterfowl flock density (aOR: 6.8, 95%CI: 3.41–14.46 and 9.17, 95%CI: 3.73–26.20, in 2015 and 2017, respectively) and higher percentage of cropping land coverage (aOR of 1.36, 95%CI: 1.10–1.69 and 1.04, 95%CI: 1.02–1.07, in 2015 and 2017, respectively). Our study highlights the application of remote sensing and clustering analysis for the identification and characterization of environmental factors in facilitating and contributing to the persistent circulation of certain subtypes of H5Nx in poultry farms in Taiwan.

the Taiwan Agriculture Research Institutional Data Access / Ethics Committee (contact via rs@tari. gov.tw) for researchers who meet the criteria for access to confidential data. Once granted for access, the minimum dataset as well as related metadata and methods required will be available to replicate all study findings reported in the article. Dataset of HPAI outbreak farms and slaughter houses are available from the Bureau of Animal and Plant Health Inspection and Quarantine (BAPHIQ) Ethics Committee (contact via yhchen0506@mail. baphiq.gov.tw) for researchers who meet the criteria for access to confidential data. Once granted for access, the minimum dataset as well as related metadata and methods required will be available to replicate all study findings reported in the article. The other datasets including wetland coverage, cropland and population density are available from Taiwan Government Open Data (data.gov.tw), which are freely downloadable.

**Funding:** The work is funded by BAPHIQ (108AS-21.3.2-BQ-B1). The funders had no role in study design, data collection and analysis, decision to publish, or preparation of the manuscript.

**Competing interests:** The authors have declared that no competing interests exist.

# Introduction

Avian influenza virus (AIV) belongs to the genus Influenza A virus of the family *Orthomyxoviridae*. The virus can infect a variety of avian species, but the low pathogenic AIVs do not usually cause explicit symptoms in poultry [1]. In contrast, the Eurasian lineage of the highly pathogenic avian influenza (HPAI) A(H5N1) viruses, with A/goose/Guangdong/96-like (GD/96) HA genes (first reported in domestic geese in southern China in 1996), have continued to circulate and cause outbreaks among poultry and wild birds in Asia, the Middle East, North America, and Africa [2,3]. Due to their rapid diversification, HPAI H5N1 viruses have been classified into clades (0 to 9) and sub-clades on the basis of their haemagglutinin (HA) gene sequence, as well as by genotypes according to their internal gene constellations [4,5].

Re-assortment events of HPAI H5N1 with low-pathogenic avian influenza (LPAI) viruses have been detected in China since 2009 in domestic and wild bird populations [3]. These events resulted in the generation of at least 13 different combinations of HA and neuraminidase (NA) subtypes, now categorized as HPAI virus, clade 2.3.4.4. Clade 2.3.4.4, appear to have first emerged as an H5 monophyletic group of viruses through multiple genetic re-assortment with the HA gene segment from HPAI H5N1 clade 2.3.4 variant and other gene segments from multiple AIV co-circulating in the domestic and wild bird populations (mainly waterfowl) [3]. The viruses from this clade are unusually promiscuous, with the propensity of circulating in the same region at the same time and also quickly spreading globally, which presents a major threat to both global poultry production and public health [6]. In particular, some subtypes, such as the H5N6, have the potential to cause severe disease in people and to adapt efficient transmission in human populations [7,8]. Epidemiological investigations of waterfowl migration and poultry trade have shown that long-distance migratory birds can play a major role in the global spread of avian influenza viruses [9]. Since 2013, this wave of panzootic outbreaks has caused tremendous socio-economic losses globally as a result of culling domestic birds, temporary bans on the export of poultry products, and the disruption of local market activities due to disease control measures such as movement restrictions.

Environmental factors have been previously considered when analyzing the spatial or temporal clustering of cases with the aim to identify spatial predictors associated with spatial clustering and to map the risk of HPAI over geographical areas [10]. These factors can be classified into different categories, including (1) farming practice and local biosecurity, (2) poultry and livestock census data, (3) anthropogenic variables, (4) socio-economic variables, (5) variables indicative of the presence or abundance of wild birds, (6) variables indicative of the presence or abundance of rivers, lakes or wetlands, (7) eco-climatic variables obtained using weather station data or remote sensing, (8) land-use and cropping variables, and finally (9) topography. While considering the relationship between environmental factors and patterns of HPAI occurrence, most of the studies mainly focused on using remote sensing to identify flooded areas or vegetation indices [11,12]. Since the presence of HPAI H5N1 has been associated with three types of variables consistently across studies and region: domestic waterfowl, several anthropogenic variables (human population density, distance to roads) and indicators of water presence [10], the use of satellite image to identify domestic water fowl flocks will be explored in this study.

In Taiwan, no HPAI viruses were detected from wild birds or domestic poultry farms prior to 2003, despite practicing high-density poultry farming [13]. However, between December 2003 and March 2004, large-scale epidemics of LPAI H5N2 occurred in poultry farms in central and southern Taiwan [14]. This LPAI H5N2 virus, containing HA and NA genes from the North America-lineage H5N2 virus and six internal genes from avian influenza A/H6N1 viruses, has been in continuous circulation in poultry farms since 2003, occasionally resulting

in large outbreaks in Taiwan [15,16]. In January 2015, Taiwan experienced another epidemic caused by HPAI H5 virus clade 2.3.4.4 [17,18]. Three subtypes of HPAI virus, including Eurasian lineage of H5N2, H5N8 and H5N3, were detected unprecedentedly during the same wave of outbreaks [18]. In 2017, a second wave of HPAI poultry farm outbreaks occurred and a new subtype, H5N6, was detected [19]. Even though all the necessary control measures were implemented during the outbreaks in Taiwan, two new subtypes of H5Nx, H5N2 and H5N8, were shown to be particularly difficult to eliminate. They continue to persist in the domestic poultry farms today (As of this submission, HPAI H5N2 is still circulating in poultry farms, although H5N8 has been declared eliminated, as reported by the Taiwan government). The co-circulation of multiple HPAI subtypes (H5N2 and H5N8) has raised questions about the environmental factors related to their persistence, which will be the focus of our current study.

## Materials and methods

### Study setting

We conducted the study in the main island of Taiwan with an area of 35,883 km$^2$ and 23 million of population, which can be further divided into 19 administrative areas, including municipal cities and counties. The island is characterized by the contrast between the eastern two-thirds, consisting mostly of rugged mountains, and the western flat plains, where most of the human population live. Agricultural land and poultry farms are located mostly in the western flat plains. The geographic extent of the main island was first subdivided into 1 or 3 km$^2$ spatial output in a gridded form based on the surveillance control zoning policy implemented in Taiwan as illustrated in the section "HPAI outbreak farms" below. Each grid box measures the indicated distance per side for data processing and analysis as described below.

### Sources of data

Six geographical datasets were used in this study and further elaborated below. All data were projected in TWD97/TM2 zone 121 and geocoded using WGS84 datum by ArcGIS, version 10.3 (ESRI, Redlands, CA, USA) for mapping and visualization. All the variables used in the analysis and their abbreviations from the six datasets are summarized in Table 1. Dataset of HPAI outbreak farms and slaughter houses are granted permission by BAPHIQ. Satellite image data used are granted permission by the ethical committee of TARI. Wetland coverage dataset are granted permission by the ethical committee of ESRI.

**Set 1: HPAI outbreak farms.** Since the first H5N2 epidemic in Taiwan in 2003, all poultry farmers are mandated to report poultry health problems or unusual increases in mortality events to local animal disease control centers (LDCCs). Data on H5Nx outbreak farms in Taiwan have been collected and compiled by the Bureau of Animal and Plant Health Inspection and Quarantine (BAPHIQ), Council of Agriculture (COA) since January 2004. Cloacal samples from sick or dead poultry and wild birds were evaluated by laboratory examination using reverse-transcriptase polymerase chain reaction (RT-PCR) and virus isolation [18]. The outbreak data were based on both the mandatory clinical disease reporting system and the nationwide active surveillance program. Based on World Organization for Animal Health (OIE) guidelines, a protection zone of 1 km in radius and a surveillance zone of 3 km in radius around the infected holdings are to be established when an outbreak is detected. The active surveillance program, complementary to the mandatory clinical reporting system, was also implemented. To ensure disease detection, the active surveillance was not only applied to the surrounding poultry farms, but also included the rendering factories and slaughter houses for unusual numbers of dead poultry. The certificate of negative results based on the laboratory examination of poultry before shipped to the slaughter houses was mandated for the counties

**Table 1. Environmental and anthropogenic predictors tested for correlation against the presence of hotspot of highly pathogenic avian influenza virus type H5Nx in Taiwan.**

| Abbreviation | Description | Transform | Remarks |
|---|---|---|---|
| **Farm-related variables** | | | |
| rchickenD | Registered chicken farm density | # registered chicken farm/grid* | Removed due to high collinearity |
| rduckD | Registered duck farm density | # registered duck farm/grid | Removed due to high collinearity |
| rgooseD | Registered goose farm density | # registered goose farm/grid | Removed due to high collinearity |
| allrD | Registered farm density | # registered farm/grid | Categorized based on Percentile (0.333, 0.667, 1) = (2, 7, 154) |
| allnwaterD | Chicken farm density | # chicken farm/grid | Removed due to high collinearity |
| allwaterD | Waterfowl farm density | # waterfowl farm/grid | Removed due to high collinearity |
| PHI | Poultry heterogeneity index | $1-\{|Pc-Pw|/Pc+Pw\}$§ | Continuous |
| **Farm biosecurity-related variables** | | | |
| nrnwaterD | Non-registered non-waterfowl farm density | # non-registered farm/grid | GAM = (19, 29, 48) |
| nrwaterD | Non-registered waterfowl farm density | # non-registered farm/grid | Percentile (0.333, 0.667, 1) = (0, 1, 506) |
| rbroilerD | Broiler chicken density (Good biosecurity) | # broiler chicken/grid | GAM = (2, 12) |
| rlayerD | Layer chicken density (medium biosecurity) | # layer chicken/grid | Percentile (0.333, 0.667, 1) = (0, 1, 84) |
| rnativeD | Native chicken density (poor biosecurity) | # native chicken/grid | Continuous |
| butcherD | Slaughter house density | # slaughter house/grid | Continuous |
| **Wildbird-related variables** | | | |
| wetlandA | % wetland coverage | Wetland coverage area/grid | Continuous |
| allrice | % rice crop coverage | Rice crop coverage area/grid | Continuous |
| allcrop | % all crop coverage (exclude rice crop) | All cropping area/grid | Continuous |
| **Anthropological variables** | | | |
| popD | Population density | Human population/grid | Categorized based on Percentile (0.333, 0.667, 1) = (371, 842) |

§Pc: number of chicken divided by number of poultry; Pw: number of waterfowl/number of poultry

*each grid is equivalent to 9 km$^2$ or 3 km x 3 km for each side.

with the highest numbers of outbreak farms during Sep.-Apr. when the migrating birds arrive in Taiwan [13].

**Set 2: Poultry farms.** Although remote sensing using optical sensor method has been shown to provide the most effective data for land-cover classification, no study has applied this technology to identify domestic waterfowl farms. In recent decades, satellite images with high spatial resolution, such as Worldview, Pleiades, RapidEye, and KompSat, makes it possible to examine its association with avian influenza outbreaks. The unprecedentedly high number of poultry farms or domestic birds being culled as a result of HPAI during 2015 urged the government to perform a detailed survey of domestic waterfowl farms. Therefore, an island-wide survey, utilizing remote satellite imaging technology, was conducted by the Taiwan Agriculture Research Institute (TARI) between August 2016 and April 2017.

The goal of this survey was to generate a complete poultry farm census dataset. Three Very High Resolution (VHR, less than 0.7m) optical satellites (Pléiades, WorldView and KompSAT) were selected as the image sources. After initial screening, a total of 109 satellite images with full coverage of the island were purchased for this purpose. These purchased images were selected based on their clarity and were generally obtained during the morning to avoid clouds. They were also processed for atmospheric or geometric correction. Each image included a composite estimate of the surface reflectance of the four spectral bands at 50 cm spatial resolution, and was used to perform an initial analysis to identify ponds using the computer program Geomatica: blue (430–550 nm), green (500–620 nm), red (590–710 nm), near infrared (NIR: 740–940 nm). Active flocks were identified by visually locating ponds with waterfowl

congregation sites in nearby fields. If the genus of waterfowl couldn't be easily distinguished by evaluating the image, final identification was verified by using aerial images from the Aerial Survey Office (ASO), on-site investigations, and Google street view photographs. The obtained information was used to map waterfowl farm locations based on the 20017 cadastral map.

The obtained poultry farm census data were spatially merged with official poultry farm registration database (OPFRD) managed by the COA. If a farm identified by remote satellite imaging didn't match with the one from OPFRD, it was classified as a non-registered flock. Due to the resolution of the satellite imaging, duck and goose flocks could not be differentiated and bird numbers could not be quantified. Therefore, the non-registered poultry flocks after ground verification could only be classified as domestic waterfowl or non-waterfowl. The information from the registered poultry farm database includes numbers and types of birds raised per holding, which can be further divided into the following categories: goose, duck, broiler chicken, layer chicken, native chicken, and turkey. Backyard poultry production for family consumption is considered negligible.

The association between intensification of the poultry sector and the risk of HPAI emergence and spread has received attention, but only few studies have attempted to quantify it. Previous studies have used abundance indicators for chicken, duck or poultry farm by using threshold values to indicate the flock size, or indicators differentiating small-scale family production from large-scale specialized farming as risk factors [20–22]. Other than the farm intensity and flock size, different poultry types also imply several epidemiological factors associated with the potential of viral transmission and viral load, should a farm become infected. These epidemiological factors include the level of investment in animal health and biosecurity measures to exclude or contain the contaminants [10]. Therefore, different variables are created to produce detailed maps of poultry farm distribution and are summarized in Table 1.

Also, to understand if the sustained transmission of H5Nx is due to the high heterogeneity of the total number of domestic waterfowl over landfowl per grid, we created the poultry heterogeneity index (PHI) described by the following equation:

$$PHI = 1 - \frac{|P_c - P_w|}{P_c + P_w}$$

Here, $P_c$ is the number of landfowl (mainly chickens) divided by the number of poultries per grid, $P_w$ is the number of waterfowl (mainly duck and goose) divided by the number of poultries per grid. The higher the value of the index, the higher the percentage of domestic waterfowl and landfowl farming in said spatial grid.

**Set 3: Slaughter houses.** Since several outbreak events were detected from slaughter houses, slaughter houses might serve as potential risk factors in spreading HPAIV due to the vehicles transporting the domestic birds. There are in total 174 slaughter houses including 116 for poultry and 58 for livestock. Their geological locations were provided by the BAPHIQ, Taiwan.

**Set 4: Crop coverage.** As indicated by previous models, duck farming is associated with rice cropping in Thailand, which implies the presence of a dense network of irrigation canals [20]. A flock with HPAI infected ducks could shed viruses into the water carried by the canals and, potentially, infect a chicken farm located downstream through untreated contaminated drinking water. This could happen even in the absence of direct contact with infected farms since the HPAI virus has been shown experimentally to persist in water for at least 17 days depending on the temperature and salinity of the water [23]. The rice crop farming database is part of the agricultural land coverage database and is available from the Taiwan government open data (TGOS) website (data.gov.tw). The proportion of area covered by rice cropping and

the coverage percentage of all-the-remaining types of cropping per grid were measured in this study.

**Set 5: Wetland coverage.** Water has long been suspected to play an important role in the persistence and spread of HPAI [20,24,25]. The proportion of area covered by lakes, rivers or floods in a 3x3 km$^2$ neighborhood of each location was determined from the wetland database kindly provided by Dr. T.S. Chen of the Endemic Species Research Institute (ESRI), COA [26]. In brief, the National Land Usage Survey is conducted every ten years, starting from 1996 by the National Land Surveying and Mapping Center, Ministry of Interior, Taiwan. The survey in 1996 used satellite imageries (2m of resolution) from Formasat-2 (formerly known as ROC-SAT-2), a decommissioned Earth observation satellite operated by the National Space Organization (NSPO), Taiwan. The second National Land Usage Survey, conducted between 2006–2008, published 103 land-use categories. The wetland database was generated by extracting digital information from Formasat-2 that was used for the second National Land Usage Survey and SPOT 1–3 obtained from Center for Space and Remote Sensing Research (CSRSR), Taiwan. Due to the potential for seasonality to affect the distribution of shallow-water wetland or stream, the images obtained during or close to the rainy season was included. The result of the classification includes 26 different categories from natural lakes and rivers to man-made ponds based on the Ramsar Convention on Wetlands in 1971.

**Set 6: Human population density.** In addition to poultry farms and environmental variables, human population density was also included as a potential predicting variable for testing. The data was also obtained from TGOS, which published the year-end population census data for each calendar year. The 3x3 km grids generated here were further spatially joined with the population census data. The smallest spatial unit of the population census data was township, which is greater than the 3x3 km grid and the border doesn't always follow the border of the grid. If the grid encompasses the border of different township, the population density of the grid will be taken from the average of each township.

## Spatial analysis

We believed that performing buffer for individual outbreak farms would introduce spatially overlapping areas due to the practice of high density poultry farming in Taiwan. Since the OIE allows each country to adopt different distances for the control zone when HPAI viral infections are identified in the poultry farms, Taiwan has elected to implement the 1km and 3 km surveillance zoning policy, while other countries, including the European Union (EU), have adopted the 3km and 10km policy instead. In order to properly choose the spatial distance for further analysis, spatial autocorrelation statistics of the numbers of outbreak farms under different sizes of fishnets were measured first by Global Moran's I. Different sized grids, from 1-10km, for the whole island of Taiwan were first created using fishnets to allow spatial clustering analysis. The statistics of Global Moran's I were measured to quantify any spatial dependence by looking at whether there are any spatial aggregation of the H5Nx outbreak farms. The Z-score cut-off of 1.96 showed highly spatial dependence at the distance ranged of 1-10km with the plateau occurring at 3km (S1 Fig). Therefore, 3km was chosen to identify local hotspots as described below.

In this study, we are interested in investigating whether the spatial clustering and the associated risk factors might be influenced by different subtypes of HPAVI or the zoning distance. Anselin's local Moran's statistics, generated by Local Moran's I spatial autocorrelation analyses (LISA), were then applied to localize the specific clusters of H5Nx outbreak farms under 3km grid of the global Moran's I results. The identified High-High (HH) clusters will be viewed as hot spots, while the identified Low-Low (LL), High-Low (HL) and Low-High (LH) clusters

will be treated as cold spots. All spatial autocorrelation analyses were performed by ArcGIS, version 10.3 (ESRI)

## Modeling

According to Tobler's first law of geography, everything is related to everything else, but near things are more related than distant things [27]. Hotspot (case) and cold-spot (control) grids were selected on the basis of the aforementioned spatial cluster analysis for further regression analysis under different covariates. Since the central region of Taiwan is primarily mountainous and most of the poultry farms are located in the coastal areas, the eligible criterion of control grids was at least one poultry farm within the defined distance of 3km fishnet for all three study periods (2015–2017). In order to avoid co-linearity of the predictors, all predictors were checked for cross correlation and only the variables with correlation coefficient <0.6 were considered for inclusion in the final model. In this study, all correlation coefficient values included for final modeling were less than 0.4. All predictor variables were individually tested for association with the case-control status of 3km grids using univariable logistic regression analysis, for each study period (S1 Table). Variables with p-values greater than 0.2 were excluded from further analysis. To take into account possible nonlinear relations, some continuous variables were categorized using either a percentile classification scheme or generalized additive model (GAM) plot. When a GAM plot shows that a variable is appropriate, it is categorized based on the curves before further analysis. When a GAM plot is not possible due to a small count, the data were reclassified based on the 33rd and 67th percentile (S2 Fig). Subsequently, multivariate logistic regression analysis was used to assess the association of independent variables with the case or control status of a 3km grid. A final model was fitted using a backward stepwise procedure and selected by the Akaike's information criterion (AIC). Risk factors with $P<0.05$ using the likelihood ratio test were considered statistically significant. Regression coefficients were converted into odds ratios (ORs; $e^{\beta}$) and their respective 95% confidence intervals (95%CIs). All analyses were performed under generalized linear models with logistic link using R software.

The risk map was then generated based on the multivariate logistic regression model, defined as:

$$P = 1/(1 + \exp(-(\beta_0 + \beta_1 X_1 + \beta_2 X_2 + \cdots + \beta_n X_n))$$

Here, P is the probability of disease outbreak, βo is a constant, {X_1,..,X_n} are the variables determined by the final model and {β_0...β_n} are the regression coefficients [25].

## H5Nx viral sequences and phylogenetic tree construction

Influenza hemagglutinin (HA) sequences of H5 viruses were downloaded from Influenza Virus Resource [28] and GISAID (https://www.gisaid.org/) databases on November 24, 2017. Duplicate strains (sequences with identical strain name) were removed. Sequences were curated to discard sequences shorter than 1500 base pairs and sequences with more than 1% ambiguous nucleotides. Host, location and date of isolation were parsed from strain names. Curated sequences were aligned using MAFFT v7.313 [29] and trimmed to coding regions.

A maximum likelihood (ML) phylogenetic tree based on all H5 sequences were first inferred using FastTree v2.19 [30]. Clade 2.3.4.4 of Goose/Guangdong-like H5 viruses were identified according to the latest nomenclature system [5]. We constructed another ML tree of clade 2.3.4.4 viruses using IQ-TREE v1.6.7 [31], with a general time reversible (GTR) + gamma substitution model. The subclades containing viruses isolated during the first-wave global spread of H5Nx were identified and subsampled based on the five geographical regions

—China, Europe, North America and North Asia. The resulting dataset containing 74 sequences were subjected to Bayesian phylogenetic framework using a discrete-state continuous time Markov chain (CTMC) model to reconstruct the spatial dispersal [32]. A GTR + gamma substitution model with an uncorrelated lognormal relaxed molecular clock and a Bayesian skyride tree prior [33] were implemented. We assumed an asymmetric transition model with Bayesian stochastic search variable selection (BSSVS) for transition parameters [34]. Markov Chain Monte Carlo (MCMC) was run for 220 million steps and sampled for every 10,000 steps using BEAST v1.8.4 [35]. Two parallel runs were conducted and combined with a burnin of 20 million for each chain using LogCombiner v1.8.4 [35]. Time of the most recent common ancestor (tMRCA) of groups of viruses were extracted from the log file with Tracer v1.6. A summarized maximum clade credibility (MCC) tree was inferred by TreeAnnotator v1.8.4, and subsequently visualized with R package ggtree [36].

## Results

### Characteristics of H5Nx positive poultry farms

Based on the outbreak events of poultry farm reported to OIE by its members through the World Animal Health Information System (WAHIS), the outbreaks due to HPAI during 2015–2018 were considered unusually high in terms of the numbers of countries with affected domestic birds, the numbers of outbreaks, and the diversity of circulating subtypes (Fig 1).

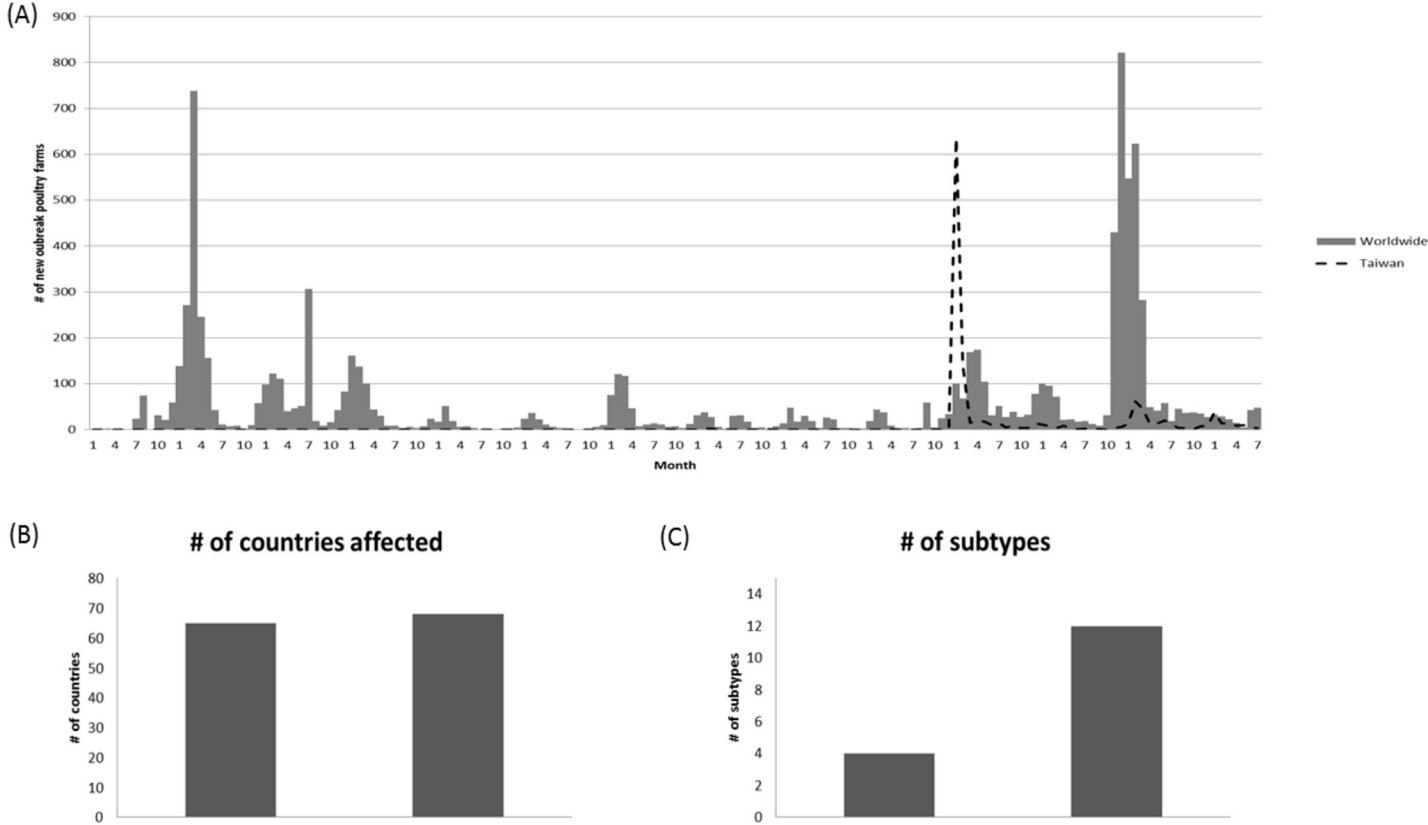

**Fig 1. Global domestic bird outbreaks due to Highly Pathogenic Influenza Virus (HPAIV) from Jan., 2005 to Jul., 2018.** (A) Numbers of new outbreak in domestic poultry farms affected by HPAIV worldwide and Taiwan from 2005 to 2018 by month. (B) Comparison of the number of countries affected by HPAIV in domestic birds between 2005–2014 and 2015–2018. (C) Comparison of the number of subtypes reported in domestic birds between 2005–2014 and 2015–2018. Data was extracted from World Animal Health Information System (WAHIS), World Organization for Animal Health (OIE).

**Table 2. Summary of the magnitude, spatial distribution and subtypes of HPAI-H5Nx outbreaks in Taiwan, 2015–2017.**

| | 2015 | 2016 | 2017 |
|---|---|---|---|
| No of outbreak farms[a]/registered farms | 1004/427 | 37/19 | 182/85 |
| Chicken | 208/111 | 18/11 | 105/58 |
| Duck | 98/29 | 4/0 | 55/15 |
| Goose | 680/278 | 14/8 | 13/8 |
| Turkey | 18/9 | 1/0 | 9/4 |
| No (%) of outbreak registered farms[b] | | | |
| Chicken (n = 6459) | 111 (1.72%) | 11 (0.19%) | 58 (0.91%) |
| Duck (n = 1683) | 29 (1.72%) | 0 (0%) | 15 (0.89%) |
| Goose (n = 748) | 278 (37.17%) | 8 (1.20%) | 8 (1.07%) |
| Turkey (n = 61) | 9 (14.75%) | 0 (0%) | 4 (6.56%) |
| No (%) of outbreak farms registered (n = 9300) | 427 (4.44%) | 19 (0.20%) | 85 (0.91%) |
| non-registered (n = 9411) | 577 (6.13%) | 18 (0.19%) | 97 (1.03%) |
| By poultry types (all farms included) | | | |
| Waterfowl (n = 9275) | 779 (8.40%) | 18 (0.19%) | 68 (0.73%) |
| Non-waterfowl (n = 9436) | 225 (2.38%) | 18 (0.19%) | 105 (1.11%) |
| Size of farms[c] | | | |
| % of <500 (n = 29) | 0 (0%) | 0 (0%) | 0 (0%) |
| % of 500–2000 (n = 194) | 23 (11.85%) | 2 (1.03%) | 4 (3.10%) |
| % of 2000–20000 (n = 5647) | 315 (5.58%) | 8 (0.14%) | 41 (0.72%) |
| % of >20000 (n = 3223) | 68 (2.11%) | 6 (0.18%) | 32 (0.99%) |
| HPAIV | 943 (93.9%) | 54 (100%) | 211 (100%) |
| H5N2 | 540 (60.7%) | 27 (50%) | 170 (80.6%) |
| H5N8 | 232 (29.8%) | 17 (31.5%) | 16 (7.6%) |
| H5N3 | 21 (2.29%) | 0 (0%) | 0 (0%) |
| H5N6 | 0 (0%) | 0 (0%) | 5 (2.4%) |
| Mixed H5N2 and | 67 (7.7%) | 2 (3.7%) | 12 (5.7%) |
| H5N8 | 2 (0.20%) | 0 (0%) | 0 (0%) |
| Mixed H5N2 and | | | |
| H5N3 | 0 (0%) | 0 (0%) | 1 (2.63%) |
| Mixed H5N2 and | | | |
| H5N6 | 81 (8.07%) | 8 (14.8%) | 7 (3.32%) |
| Non-subtyping | | | |

[a]total outbreaks events excluded 28, 17 and 29 events from slaughter house in 2015, 2016 and 2017, respectively.

[b]Among 9,300 registered farms, 349 do not find poultry type information and was classified as missing data.

[c]only can be calculated from the data of the registered farms and only 9,093 farms can find the information of poultry numbers.

The basic characteristics of the poultry farms that were confirmed to be H5Nx positive in Taiwan from January 2015 through December 2017 are shown in Table 2. The outbreak data were collected based on both the mandatory clinical disease reporting system and the nationwide active surveillance program as illustrated in the Materials and Methods section. Outbreak events identified through active surveillance programs, such as those from slaughter houses, can be difficult to trace back to the original farms and are thus excluded from further analyses. Annual outbreak events from 2015 to 2017, identified by slaughter house surveillance program, were 27 (2.62%), 54 (31.5%) and 29 (13.7%), respectively. After excluding the outbreak events identified from the slaughter house, 1,004 (5.37%), 37 (0.2%) and 182 (0.97%) poultry

farms with laboratory-confirmed H5Nx were detected among 18,711 poultry farms (including both registered and non-registered flocks) in Taiwan for 2015, 2016 and 2017, respectively. Turkey farms, despite suffering the high percent of outbreaks, represent a relatively small portion of poultry farms in Taiwan and will therefore be excluded. Three major types of poultry farms including chicken, duck and goose will be discussed here. Throughout these three consecutive years, domestic goose farms had the highest percentages affected by HPAI among three types of poultry farms with 37.17%, 1.20% and 1.07%, respectively. Nevertheless, the duck farms had the lowest attack rates. Interestingly, if taking into account of all poultry farms including non-registered flocks, the highest percentage of farms being affected shifts from domestic waterfowl to domestic non-waterfowl farms, with 8.4% and 2.38% attack rate in 2015, to 0.73% and 1.11% rate in 2017, for both types of poultry farms, respectively.

The spatial and temporal distributions of the infected farms between 2015–2017 are shown in Figs 2 and 3, respectively. The major poultry farms affected by H5Nx are distributed along the coastal areas of central and southern Taiwan, particularly in Yun-Lin (YL) and Ping-Tung (PT) counties (Fig 2B). However, it was noted that the areas with the highest intensity of poultry farming were primarily located in YL and Chang-Hwa (CH) counties, not PT county (Fig 2A).

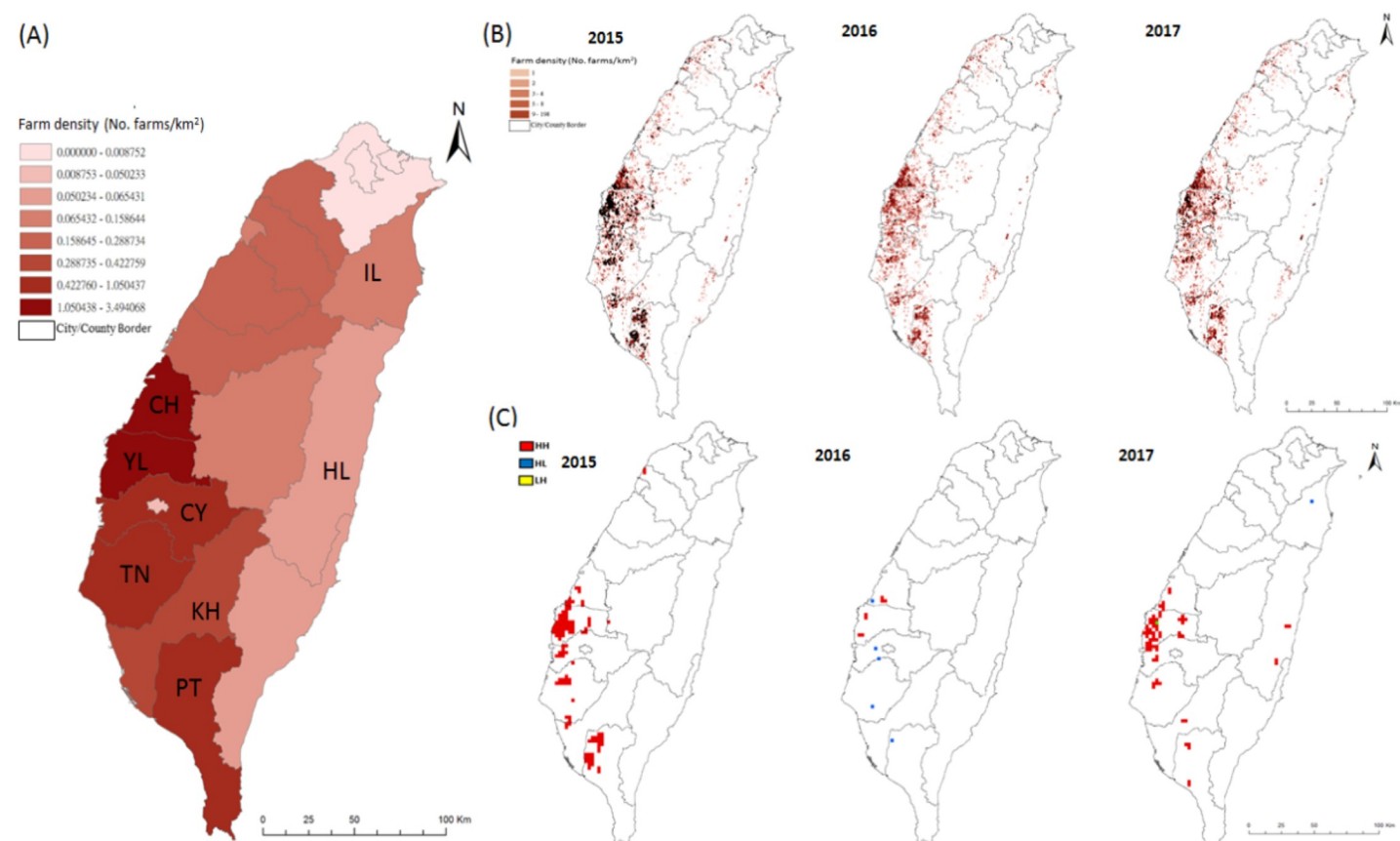

**Fig 2. Spatial distribution of poultry farm density and the presence of Highly Pathogenic Avian Influenza (HPAI) H5Nx in poultry farms and identified hotspot area in Taiwan from 2015 to 2017.** (A) Poultry farm density calculated by numbers of farms per square kilometer for each city/county. The drawn border of the polygons indicated 19 administrative areas, including 10 counties and 9 cities. Alphabetic letter indicated the name of county: CH-Chang-Hwa, YL-Yun-Lin, CY-Chia-Yi, TN-Tai-Nan, KH-Kao-Hsiung, PT-Ping-Tung and HL-Hua-Lien. (B) Distribution of poultry farm plotted by numbers of farms in 3x3 grid with 3 kilometer for each side of the grid. Each solid black dot indicated one outbreak poultry farm. (C) Spatial clustering of HPAI outbreak farms under 3x3 km$^2$ size of grid identified by Local Moran's I spatial autocorrelation analyses from 2015 to 2017. HH, indicated high-high hotspot area; HL indicated high-low hotspot area; LH indicated low-high hotspot area.

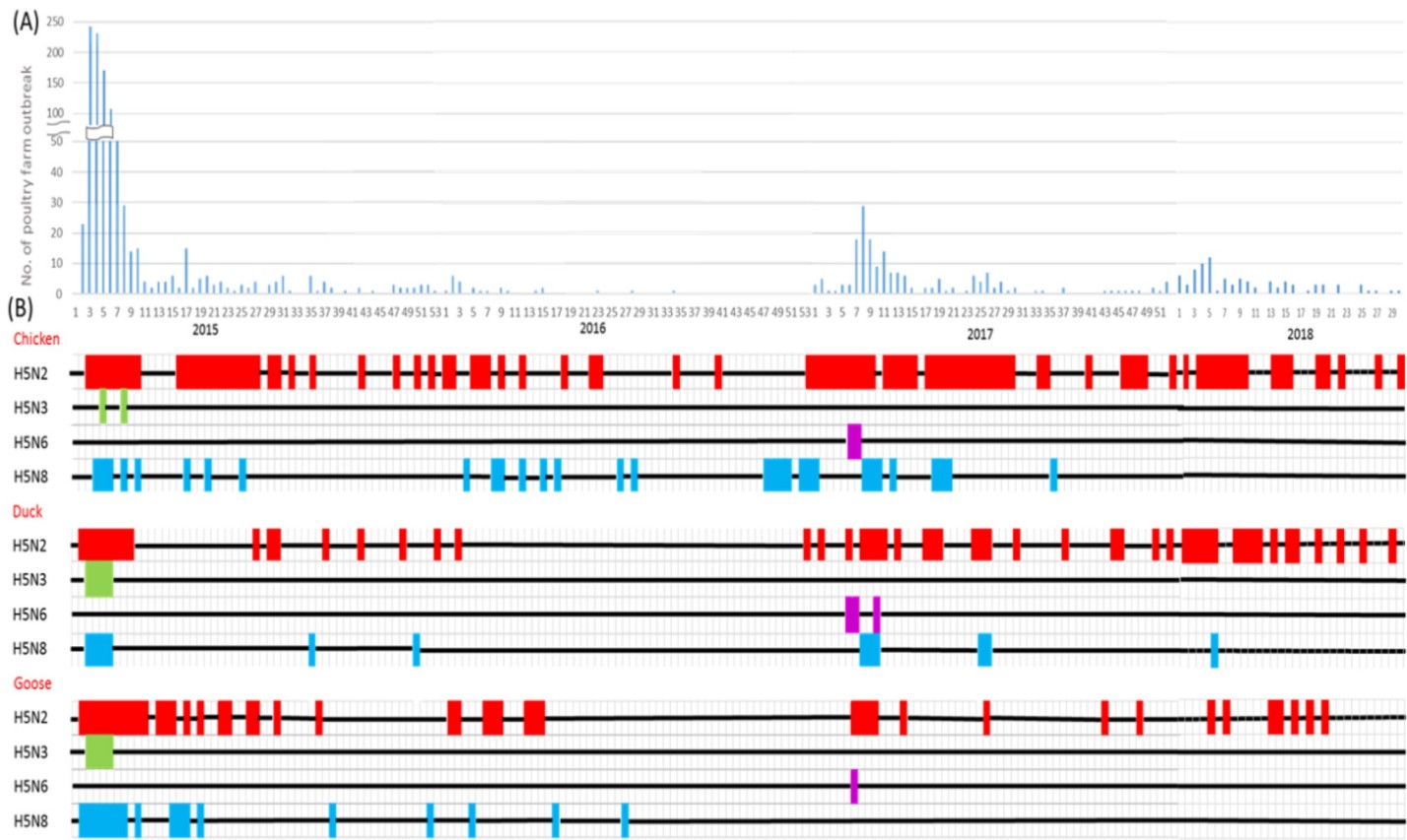

**Fig 3. Temporal distribution of Highly Pathogenic Avian Influenza (HPAI) H5Nx in poultry farms by subtypes in Taiwan from Jan., 2015 to Jul., 2018.** (A) Epidemic curve of HPAI outbreak farms by week during 2015–2018. (B) Timeline of the appearance of HPAI by poultry type and H5Nx virus subtype. The color code for each subtype is H5N2 (red), H5N3 (green), H5N6 (purple), and H5N8 (cyane). The timeline plotted by week corresponded to the time axis plotted in (A).

The HPAI-affected poultry farm outbreaks in Taiwan during 2015–2017 came in two waves: the first wave started in January, 2015 caused by the newly introduced HPAI H5N2, H5N8 and H5N3 to the island and the second wave was caused by the other newly introduced H5N6, which was particularly apparent after February, 2017 (Fig 3). The peak months of the H5Nx-confirmed poultry farms were detected during January and February (S3 Fig). According-ing to the subtypes of H5Nx identified from the poultry farms, H5N3 was only detected in 2015 and H5N6 only in 2017. However, H5N2 and H5N8 have been continuously detected among the poultry farms for three consecutive years since their first arrival in 2015, although different intensity was noticed (Fig 3). Specifically, H5N2 was persistently detected in the chickens, ducks and geese in Taiwan in three years from 2015 to 2017. However, H5N8 could only be detected in geese in 2015–2016, but not after the summer of 2016 and 2017.

## Spatial clustering identification

Hot spot analysis using LISA under a 3x3 km grid size, identified multiple areas with signifi-cant spatial clustering of H5Nx outbreak farms. They are mainly located along the west coast of Taiwan between 2015 and 2017 (Fig 2C). Further calculation of the percentage of outbreak farms covered by the hot spot areas found that, around 75% of outbreak farms were located in the hot spot areas, with 75.7% and 76.37% for the 2015 and 2017 periods, respectively, with the exception that only 24.32% was covered in 2016.

Further examination of the hot spots related to H5Nx outbreaks by different subtypes based on a 3x3 km grid analysis, yielded results that showed the spatial clustering of H5N3 during 2015 was mainly located in PT county, which is highly overlapped with the hot spot areas of H5N2 and H5N8 (S4 Fig). In contrast, the spatial clustering of H5N6 in 2017 were mainly located in YL, CY and HL counties and did not overlap with any hot spot area affected by H5N2 and H5N8 (S5 Fig). Interestingly, neither H5N3 nor H5N6 circulated persistently in poultry farms and the outbreak farms caused by both subtypes were quickly under control. In summary, the spatial clustering analysis suggests that the H5Nx-confirmed outbreak poultry farms in Taiwan were highly spatial-clustered, which might be attributed to local environmental factors. Furthermore, the geographical locations of the hot spots for H5N6-affected poultry farms were different from those for H5N2- and H5N8-affected farms, implying different environmental factors drove such clustering.

## Risk factor analysis

In order to understand what environmental factors were associated with the spatial clustering of HPAI(+) outbreak farms, we used spatial-based case-control study design and applied multivariate stepwise logistic regression to compare the hot spot and non-hot spot areas identified between 2015–2017. The list of environmental factors used in this study and their abbreviations are summarized in Table 1. We first performed the comparison for different calendar years and the results are summarized in Table 3. Since there weren't enough identified hotspot areas in 2016, the estimation was unstable. Nonetheless, four risk factors consistently showed strong association with the identified hotspots in 2015 and 2017. These include nrwaterD, allrD_high, PHI and allcrop. High non-registered waterfowl flocks density (nrwaterD) showed significantly correlated with hotspot area with adjusted odds ratio (aOR) 6.8 (95% CI: 3.41–

**Table 3. Multivariate logistic regression modeling results after stepwise selection comparing the hot zone and non-hot zone areas of HPAIV-confirmed outbreak farms based on 3km local spatial clustering analysis during 2015–2017, Taiwan.**

| | 2015 | | | | 2016 | | | | 2017 | | | |
|---|---|---|---|---|---|---|---|---|---|---|---|---|
| | Estimate | aOR※ | 95% CI$ | p-value | Estimate | aOR※ | 95% CI | p-value | Estimate | aOR※ | 95% CI | p-value |
| nrwaterD | | | | | | | | | | | | |
| medium | 0.87 | 2.38 | 0.90–6.18 | 0.07 | | | | | 1.71 | 5.51 | 1.81–17.90 | <0.01** |
| high | 1.92 | 6.80 | 3.41–14.46 | <0.001*** | | | | | 2.22 | 9.17 | 3.73–26.20 | <0.001*** |
| allrD | | | | | | | | | | | | |
| medium | 1.47 | 4.35 | 1.34–19.50 | 0.03* | | | | | 1.62 | 5.07 | 1.28–33.74 | 0.04* |
| high | 2.86 | 17.46 | 5.91–74.86 | <0.001*** | | | | | 2.11 | 8.23 | 2.12–54.86 | 0.01** |
| PHI | 2.51 | 12.28 | 5.02–31.14 | <0.001*** | 2.10 | 8.13 | 0.65–71.99 | 0.07 | 1.02 | 2.79 | 1.00–7.69 | 0.05* |
| allcrop | 0.31 | 1.36 | 1.10–1.69 | <0.01** | | | | | 0.04 | 1.04 | 1.02–1.07 | <0.001*** |
| rnativeD | | | | | 0.11 | 1.11 | 1.01–1.21 | 0.01* | 0.04 | 1.04 | 0.99–1.10 | 0.14 |
| popD | | | | | | | | | | | | |
| medium | 0.32 | 1.37 | 0.72–2.65 | 0.34 | -0.58 | 0.56 | 0.10–3.29 | 0.50 | -0.43 | 0.65 | 0.35–1.21 | 0.18 |
| high | -0.38 | 0.69 | 0.32–1.46 | 0.33 | -16.90 | 0.00 | NA | 0.99 | -2.02 | 0.13 | 0.04–0.37 | <0.001*** |
| butcherD | 0.42 | 1.52 | 0.89–2.60 | 0.12 | | | | | | | | |
| allrice | -0.20 | 0.81 | 0.69–0.95 | 0.01* | 0.03 | 1.03 | 1.00–1.07 | 0.07 | | | | |

※aOR: adjusted odds ratio

$CI: confidence interval.

*p<0.05

**p<0.01

***p<0.001.

14.46) and 9.17 (95% CI: 3.73–26.20) in 2015 and 2017, respectively. High poultry farm density (allrD_high) was shown to give 17.46-times (95%CI: 5.91–74.86) and 8.23-times (95% CI: 2.12–54.86) higher risk of becoming a hotspot of HPAIV for 2015 and 2017, respectively. High poultry heterogeneity index (PHI) also showed consistently strong correlation with the hotspot areas, with aOR of 12.28 (95%CI: 5.02–31.14) and 2.79 (95%CI: 1.00–7.69) for 2015 and 2017, respectively. Higher cropping land coverage percentage (allcrop) was also consistently correlated with the spatial clustering in 2015 and 2017, with aOR of 1.36 (95%CI: 1.10–1.69) and 1.04 (95%CI: 1.02–1.07), respectively.

To avoid instability in the estimates for 2016 and in compliance with the two waves of epidemics (Fig 3), we performed another analysis based on aggregating the hot spots identified during 2015–2016 (S2 and S3 Tables). The results showed that three variables, including nrwaterD, allrD_high and PHI, remained strong factors associated with the hotspots during the first and second waves of the HPAI epidemics. The respective aOR for three variables were 14.52 (95% CI:6.0–43.44), 23.58 (95% CI:6.85–148.53) and 9.55 (95% CI:3.95–23.95) (S2 Table).

Since these three factors consistently showed strong association with the HPAIV hotspot areas during 2015 and 2017, they are likely associated with the persistent circulation of HPAIV among the poultry farms in Taiwan, particularly H5N8 and H5N2. To further prove this point, multivariate stepwise logistic regression was applied to identify any risk factors associated with the hot spot areas of H5N3 during 2015 and H5N6 during 2017. The results showed that high PHI was a significant factor associated with hotspot areas of H5N3 and was also the only factor found in a similar analysis of H5N2 and H5N8 (S4 Table). The other biosecurity-related factors, including high density of layer and native chicken farms were found to be significantly associated with the spatial clustering of H5N3. However, they were not found to be associated with the clustering of H5N2 and H5N8. Furthermore, none was identified as an important ecological factor associated with spatial clustering of H5N6 (S4 Table).

### Risk map and validation

We further generated risk maps, utilizing the estimates based on the regression models from the 2015 (Fig 4A) and 2017 (Fig 4B) calendar years, or from the first wave of 2015–2016 (Fig 4C). The predictions for high-risk areas were further validated using the poultry farm outbreak events confirmed from January to July 2018. The results showed that 87%, 0%, 55% of the poultry farms affected by HPAI in 2018 fell into the areas predicted with risks higher than 0.7 in Fig 4A, 4B and 4C, respectively.

### Discussion

Since the emergence of new subtypes of HPAI H5Nx viruses in China, the diseases caused by the viruses have spread widely in countries across Asia, Europe, Africa, and the Middle East [3]. Outbreaks of HPAI H5Nx virus infection in poultry farms have resulted in a high number of affected animals, losses in domestic and international trade of poultry products with significant socioeconomic impacts and public health consequences. The continuing emergence and transmission of HPAI H5Nx has raised global concerns regarding potential threats to human health due to the reports of human infection and death [37,38]. Despite the fact that HPAI H5Nx viruses continue to be a major threat for animal and public health worldwide, the risk factors associated with clustering and persistence in H5Nx-infected poultry farms have remained largely unexplored. The multiple subtypes of HPAI H5Nx epidemics in Taiwan between 2015–2017 provide a good opportunity for the study of risk factors associated with the persistence and spread of H5Nx in clustering areas, particularly those of H5N8 and H5N2.

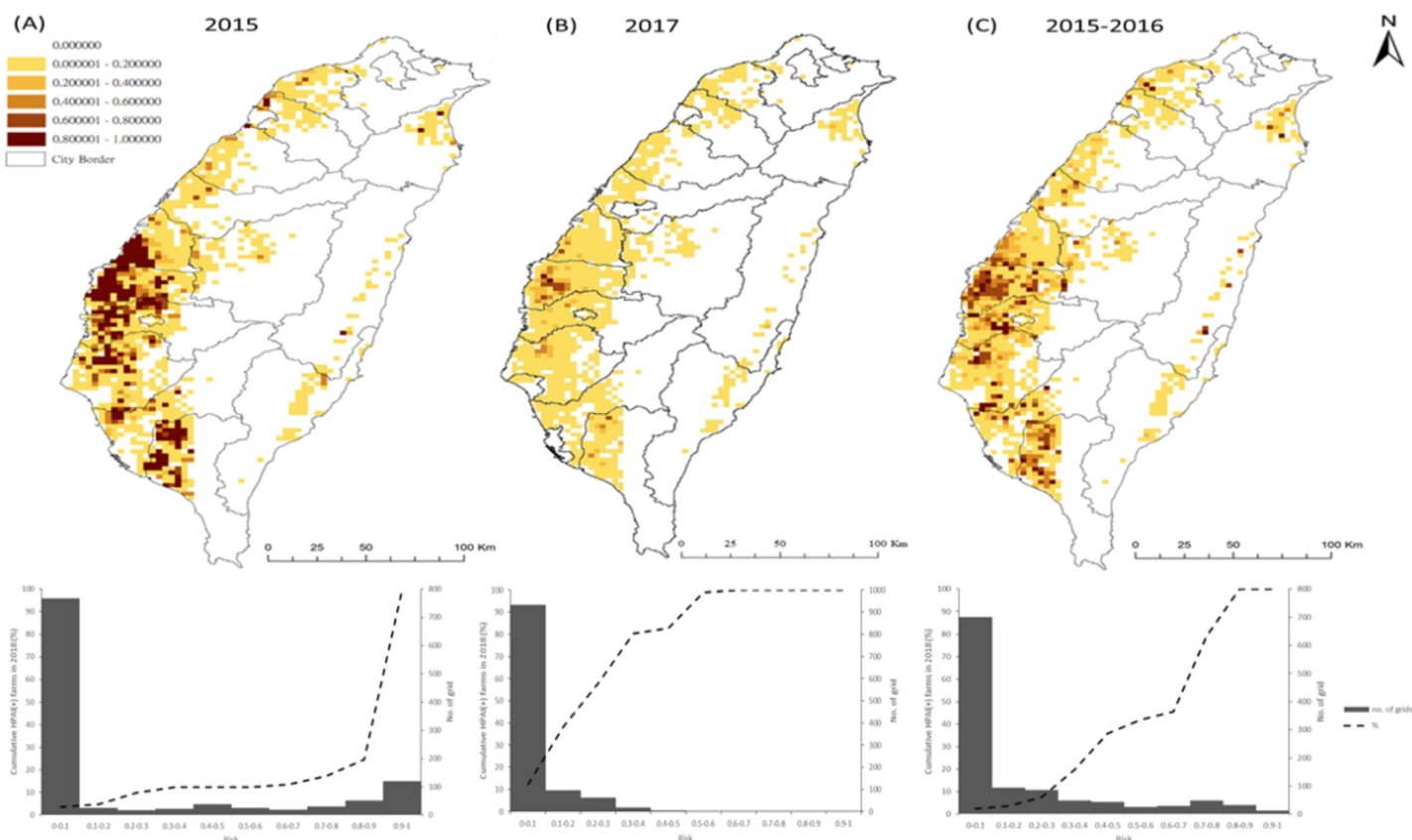

**Fig 4.** Risk maps and prediction validation based on different regression modeling on 2015 (A), 2017 (B) and 2015–2016 hot-spot areas. (Top panel) The distribution of probability of becoming HPAI-affected hot spots. (Bottom panel) Cumulative percentage of HPAI-affected outbreak poultry farms confirmed during Jan-Jul, 2018 over ranges of risk.

Utilizing satellite imaging technology, our study identified four risk factors consistently associated with the clustering of HPAIV-affected poultry farms during 2015–2017, including non-registered waterfowl flock density (nrwaterD), high poultry farming density (allrD), PHI and cropping density; although other risk factors also contributed to the clustering of HPAIV in different years. Fig 5B–5E shows individual density maps of each variable. The RGB composite plot in Fig 5A depicts the overlapping distribution of the three predictive variables. The high poultry farm density in the vicinity, in particular the high heterogeneity of farm types (PHI), created a good environment for HPAI virus to transmit from domestic waterfowl, the reservoir, to chickens, which makes the control measures implemented so ineffective, and allowed the HPAI viruses (H5N8 and H5N2) continuing to circulate. The situation exacerbated with the surrounding by non-registered waterfowl flock density. Since the policy stance of HPAI control measures in Taiwan is to not vaccinate the poultry population, the results from our current study provide important insights into the mechanisms allowing the virus to persist in poultry farms. Importantly, when calculating population attributable risk, 12.8% and 56% decrease in risk can be achieved through the reduction in poultry farm density from high to medium and low density, respectively.

Satellite technology has been applied with some success to predictive modeling of infectious diseases, including influenza, in recent decades [11]. While considering the relationship between environmental factors and patterns of infectious disease occurrence, most of the studies mainly focused on using remote sensing to identify flooded areas or vegetation indices

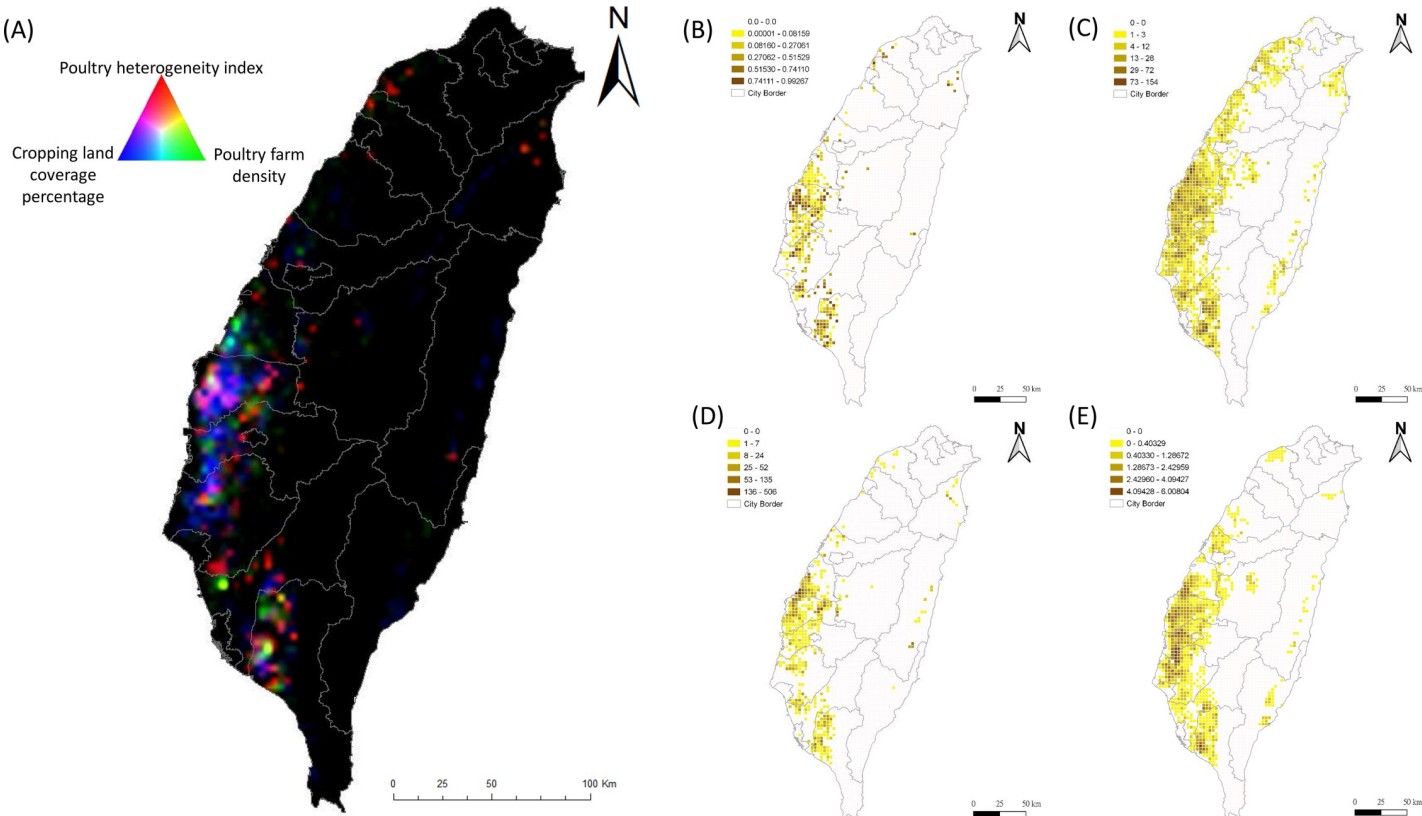

**Fig 5. Spatial distribution of predictor variables responsible for the persistence of H5N2 and H5N8 HPAI H5-subtypes among poultry farms in Taiwan.** (A) RGB composite plot of the distribution of three predictor variables: poultry heterogeneity index (domestic anseriformes to galliformes density (red), poultry farm density (green), and cropping land coverage percentage (blue). Dark areas correspond to low values and light areas to high values in all 3 predictors. Black indicated no poultry farm. Density map of poultry heterogeneity index (B), all registered poultry farms (C), non-registered waterfowl flocks (D), and non-rice cropping area coverage (E) in Taiwan. Density is expressed as numbers of flocks per 3x3km grid.

[20,25,39–41]. We explored the poultry farm density, in particularly the non-registered waterfowl flocks, using satellite imaging in the current study. High non-registered waterfowl farm density identified through satellite imaging reported 6.8-times higher risk of having outbreak farm clustering (Table 3). Although remote-sensing technology is currently just a research tool, the satellite data are becoming increasingly accurate through different ground verification methods. It is also reasonable to assume that the environmental and ecological conditions that lead to irregular outbreaks would change over time. The extensive survey of poultry farms through satellite imaging in the current study was performed only once during the autumn period in 2016 and the potential bias of applying this one-time survey to all three years (2015–2017) cannot be excluded. However, the strong association between high non-registered waterfowl farm density and the clustering of HAPI-affected poultry farms, highlights tremendous potential for using satellite imaging for future research in avian influenza outbreaks. A dynamic and continuous collection of high resolution satellite imagery could be considered in the future.

A recent review of spatial risk factors associated with H5N1 HPAI presence through different studies highlighted domestic poultry farm density (either duck or chicken), anthropogenic variables (human population density as a proxy of poultry flock density), and indices of water presence (distance to rivers and proportion of land occupied by water) to be the most consistent risk factor across studies, countries and scales [10]. Although their aOR are different from

2015 to 2017, the confidence intervals are overlapping. Our study further identified the PHI (Ivofmix) index as an important risk factor for the high-clustering of HPAI outbreaks throughout the three consecutive years (Table 3). Similarly, the emergence and ongoing transmission of avian influenza A(H7N9) virus in China since 2013 also demonstrated a geographic shift to the areas with high chicken densities and high chicken-to-duck ratios [42]. Recent findings have shown H5Nx viruses spreading among domestic ducks and chickens with different transmissibility and pathogenicity, with H5N6 found to be highly pathogenic in both chickens and ducks and H5N8 being lethal in chickens but causing no remarkable clinical illness in ducks [43–45]. However, no similar pathogenic study was performed for H5N3 or H5N2. This was consistent with HPAI H5N1, which indicated that domestic ducks play an important role as the reservoir for sustained transmission of the influenza virus among the poultry farms [46,47]. Ducks are inherently more resistant to diseases from HPAI viral infections than gallinaceous poultry, because they have retained the IFN-inducing RIG-I family of genes, lost from the jungle fowl that are the precursors of domestic chickens [48]. Goose seems remain susceptible to HPAI infection although it retains the IFN-inducing RIG-I gene [49–51]. Other virological factors, different between H5N3/H5N6 and H5N2/H5N8, such as transmissibility, pathogenicity and persistence of virus in waterfowls or other poultries, may play a major role on the persistence of H5N2/H5N8, but not H5N3/H5N6. Since the ducks usually don't present any clinical symptoms after being infected by H5N8, the high domestic waterfowl to landfowl density could provide a good opportunity for the H5N8 virus to persistently circulate among waterfowl farms and spread to chicken farms when both types of farms are spatially close.

Waterfowls play an important role as a major influenza virus reservoir, harboring genes that might contribute to HPAIVs with the potential to cause severe diseases in gallinaceous poultry [52]. Several studies have suggested that migratory birds played an important role in long-distance transmission of HPAI H5Nx along the flyways [7,9]. The evidences supporting migratory birds play an important role in introducing novel HPAI subtypes into Taiwan was provided as the following. First, all four subtypes of H5Nx identified during 2015–2017 shared the same HA gene lineage derived from A/goose/Guangdong/1/96-like (Gs/GD) HPAI H5, which has never been identified before in Taiwan [18,19]. Phylogenetic analyses, based on influenza HA genes, demonstrated that HPAI H5Nx viruses isolated in poultry farms during the first wave in Taiwan were closely related to two Japanese isolates identified from wild birds in late 2014 (S6A and S6B Fig). Along with other geographical areas involved in the same wave of global H5Nx expansion, HPAI viruses, discovered in Korea and Japan, could be the sources of viruses introduced to Taiwan in winter 2014 (S6C Fig). Secondly, Monitoring HPAI H5N2 genetic changes based on active virological surveillance system in Taiwan has revealed that re-assortment of internal genes with multiple LPAI viruses from wild birds has occurred, although the HA and NA genes remain the same (BAPHIQ website ai.gov.tw). In particular, a new re-assorted HPAI H5N5 with HA gene belonging to clade 2.3.4.4 descendent with Taiwan local strains, was identified since September, 2019. Thirdly, our viral sequencing data from the field collections during 2019 provided the directly scientific support of gene exchange between the domestic poultry and wild birds [53]. Fourth, avian influenza virus monitoring in migratory birds has been performed in Taiwan since 1998 and was recently expanded to include the dead wild birds collected by the citizen scientists [13]. The results also showed the avian influenza viruses can be isolated from the domestic wild birds [54] (https://ai.gov.tw/index.php?id= 2017). The reasons of the elusive transmission mechanism between migratory birds and domestic poultry are mainly due to the low sampling rate of wild birds from the surveillance system implemented and different susceptibility of wild birds infected by different subtypes of HPAI. Although the possibility that HPAI H5Nx were introduced through illegal trade of live birds and animal products cannot be completely excluded, our study highlighted the need for

more sampling from migratory birds to reveal the transmission pattern by "bridging species" and evolution of local HPAI [55,56].

In our study, wetland or paddy rice coverage density was not significantly associated with the high-clustering of HPAI-affected poultry farms. In contrast, the cropping density remains an important risk factor throughout the period of 2015–2017, which might indicate the potential role of non-migratory wild birds in the transmission of the HPAI virus by bridging poultry farms and migrating birds from the wetlands. We have obtained the wild bird dataset, collected through citizen scientists in Taiwan, and the dynamic association of non-migratory wild birds with poultry farms is currently under investigation. The recent North American experience with HPAI H5Nx suggests that these viruses are not well fitted to persist in wild bird populations and will likely disappear without an endemic poultry source [57]. Increasing the biosecurity of poultry farms will be necessary to contain the further movement of HPAI between domestic poultry and wild birds.

After the first isolation of HPAI virus on January 5, 2015, the Central Emergency Operation Center (CEOC) was established and control strategies were immediately implemented. The control measures included four different components: (1) both passive and active surveillance and diagnosis, (2) culling and depopulation, (3) biosecurity and restriction on poultry movement, and (4) education [58,59]. Positive flocks or flocks within 3 km of an affected farm where the death rate was greater than 10% were destroyed without delay. Infected premises and equipment were cleaned and disinfected. Furthermore, movement of poultry and their products was restricted within a 1–3 km radius around an affected farm. With continuing reporting and confirmation of HPAI-affected farms, poultry slaughtering and marketing were suspended for four days starting on Jan 24, 2015. Marketing of poultry meat was suspended again from February 17–24, 2017, immediately after the confirmation of poultry affected by another novel HPAI subtype (H5N6). After necessary control measures were implemented, HPAI H5N3 was officially eliminated from Taiwan on May 22, 2015 and H5N6 was eliminated on May 5, 2017. However, due to the continuous circulation of HPAI H5N2 and H5N8, the active surveillance was not only applied to the surrounding poultry farms, but also included the rendering factories and slaughter houses for unusual numbers of dead poultry to ensure disease detection. The certificate of negative results based on the laboratory examination of poultry before shipped to the slaughter houses was mandated for the counties with the highest numbers of outbreak farms during Nov.-Mar. when the migrating birds arrive in Taiwan. Finally, it took three years for H5N8 to control and was officially eliminated after Feb. 22, 2018. The reasons that HPAI H5N2 are particularly difficult to eliminate are probably due to at least two factors. First, domestic ducks with low pathogenicity to HPAI play a role in preserving the viruses. High HPAI H5 seropositivity in duck farms, based on the results of avian influenza sero-surveillance system implemented by BAPHIQ, confirmed that domestic ducks play an important role as the reservoir for sustained transmission and influenza virus reassortment events among poultry farms. Waterfowl play an important role as an influenza virus reservoir, harboring genes that might contribute to HPAIVs. These viruses have the potential to be introduced into landfowl holdings and cause severe disease in gallinaceous poultry through direct contact between poultry and infected wild birds or through indirect contact with fomites (e.g., water, feed bedding material, boots, and wheels of vehicles) contaminated with wild bird feces [9,52]. Second, the co-circulation of LPAI and HPAI H5N2 at the same time among the poultry farms. After four years of control efforts, this new subtype of HPAI H5N2 was shown to be particularly difficult to eliminate and continues to persist, together with LPAI Mexican-lineage H5N2, in domestic poultry farms. Taiwan has implemented intense surveillance and control measures since 2015 and has reported more poultry outbreak events to OIE than any other country (Fig 6).

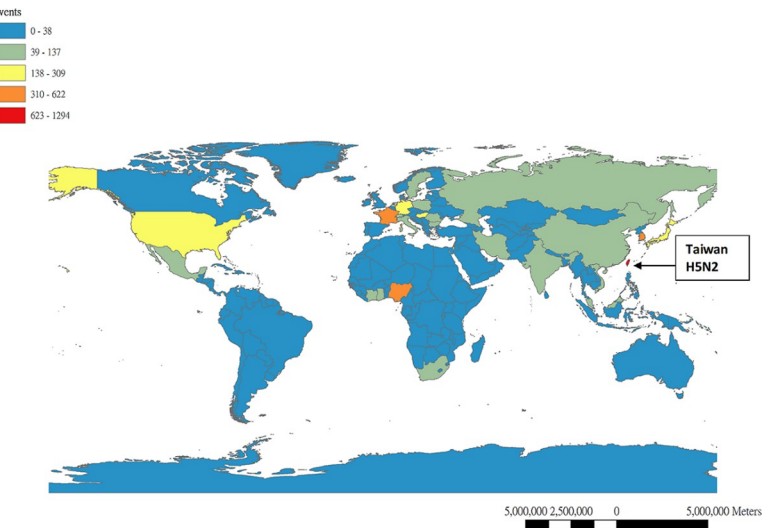

**Fig 6. Global distribution of HPAI poultry farm outbreaks reported to OIE from Jan. 1, 2009-Jun. 30, 2019.**

Our current study cannot exclude the possibility of under-reporting of HPAI virus-infected farms in non-hot spot areas, which could be attributed to the transmission of the HPAI virus through direct contacts between a poultry and an infected wild bird or an indirect contact with materials (e.g., water, feed bedding material, boots, and wheels of vehicles) contaminated with wild-bird feces. These have been previously considered as the most likely route of virus introduction into poultry holdings [9]. However, the likelihood should be low since the compensation policy in Taiwan is implemented to encourage farmers that encounter unusual poultry mortality to report to the LDCC voluntarily. The lower number of outbreak farms identified during 2016 might be due to the recovery period resulting from the large outbreaks occurred in previous year. Also, our study cannot exclude the likelihood of chicken farms as the alternative potential source of virus transmission in Taiwan due to the persistence of lowly pathogenic domestic H5N2 virus circulating in the poultry farm, which might allow some replication of HPAI H5Nx viruses without clinical signs. The speculation is supported by the high sero-positive rate in layer chicken farms, based on the results of avian influenza active serological surveillance (https://ai.gov.tw/index.php?id=970). Due to the lack of biosecurity information of each farm, the methods in which HPAI viruses transmit among the farms need further investigation. Also, tracking the genetic variation of HPAI H5Nx has the potential for further genotyping based on eight different gene segments to reveal different genotypes in transmission dynamics and their interaction with different environment factors. The wide range of landfowl permissive to infection and sustainable transmissibility with the wild bird adapted 2.3.4.4 virus, highlights the potential of this virus acquiring mutation leading to host range adaptation [60]. However, the only available data in this study is the subtyping of each outbreak farms and the detail virological sequences from each outbreak farms are not available. The integration of phylogeographical analysis and environmental factors can be done in the future study.

In summary, our study suggested that the synergistic effects of high density of poultry farms, high poultry heterogeneity index, and cropping land contributed to the spatial clustering and persistence of H5N2 and H5N8 between 2015–2017. As the chance of contact between domestic poultries and wild bird increases, the increased risk of high transmission capacity between domestic poultry and humans resulting from re-assortment highlight the importance

of stronger bio-security measurements required to face the emergence and spread of HPAI H5 subtype virus clade 2.3.4.4.

## Supporting information

**S1 Table. Univariate logistic regression modeling results after stepwise selection comparing the hot zone and non-hot zone areas of HPAIV-confirmed outbreak farms based on 3km local spatial clustering analysis during 2015–2017, Taiwan.**
(DOCX)

**S2 Table. Univariate logistic regression modeling results after stepwise selection comparing the hot zone and non-hot zone areas of HPAIV-confirmed outbreak farms based on 3km local spatial clustering analysis during two epidemic waves of 2015–2016 and 2017, Taiwan.**
(DOCX)

**S3 Table. Multivariate logistic regression modeling results after stepwise selection comparing the hot zone and non-hot zone areas of HPAIV-confirmed outbreak farms based on 3km local spatial clustering analysis during two epidemic waves of 2015–2016 and 2017, Taiwan.**
(DOCX)

**S4 Table. Multivariate logistic regression modeling results after stepwise selection comparing the hot zone and non-hot zone areas of HPAIV-confirmed outbreak farms based on 3km local spatial clustering analysis of H5N3 and H5N6 in 2015 and 2017, respectively.**
(DOCX)

**S1 Fig. Spatial autocorrelation (Global Moran's I) of outbreak farms due to HPAI viruses by distance.** A peak Z score at 3km suggests that spatial processes exist at this distance to produce pronounced spatial clustering.
(DOCX)

**S2 Fig. Plots of Generalized Additive Model (GAM) with the identified hotspots as the dependent variable.** (a)-(h) showed different predictors on HPAI hotspot areas. (a) All registered farm density (allrD) showed non-linear relationship with dependent variable. However, since the sparse data points at high farm density, allrD was trichotomized based on 33 and 67 percentile. (b) Non-registered non-waterfowl flock density (nrnwaterD) showed non-linear relationship with dependent variable and was trichotomized at 19 and 29. (c) Non-registered waterfowl flock density (nrwaterD) showed non-linear relationship with dependent variable. However, since the sparse data points at high farm density, allrD was trichotomized based on 33 and 67 percentile. (d) Registered broiler chicken farm density (rbroilerD)) showed non-linear relationship with dependent variable and was trichotomized at 2, 12. (e) Registered layer chicken farm density (rlayerD) showed non-linear relationship with dependent variable. However, since the sparse data points at high farm density, allrD was trichotomized based on 33 and 67 percentile. (f) Registered native chicken farm density (rnativeD) showed linear relationship with dependent variable. (g) Butcher house density (rbutcherD) showed linear relationship with dependent variable. (h) Population density (popD104)) showed non-linear relationship with dependent variable. However, since the sparse data points at high farm density, allrD was trichotomized based on 33 and 67 percentile.
(DOCX)

**S3 Fig.** Temporal distribution of highly pathogenic avian influenza (HPAI) H5Nx in poultry farms by poultry types, (A) chicken, (B) duck, (C) goose, in Taiwan from 2015 to 2017. (DOCX)

**S4 Fig. Spatial distribution of HPAI hotspot outbreak farms by different HPAI subtypes under 3x3 km grid identified by Local Moran's I spatial autocorrelation analyses during 2015.** (a) The high-high (HH) hotspot areas for outbreak farms by H5N2 and H5N8; (b) the HH areas for outbreak farms by H5N3; (c) Merge of HH areas of H5N2/H5N8 and H5N3. The overlapping hotspots colored in yellow. (DOCX)

**S5 Fig. Spatial distribution of HPAI hotspot outbreak farms by different HPAI subtypes under 3x3 km grid identified by Local Moran's I spatial autocorrelation analyses during 2017.** (a) The high-high (HH) hotspot areas for outbreak farms by H5N2 and H5N8; (b) the HH areas for outbreak farms by H5N6; (c) Merge of HH areas of H5N2/H5N8 and H5N6. The overlapping hotspots colored in yellow. (DOCX)

**S6 Fig. Molecular dating of the global dissemination of clade 2.3.4.4 influenza A viruses during 2014–2015.** (A) Maximum likelihood phylogeny of clade 2.3.4.4 H5 viruses based on the hemagglutinin (HA) gene. The two clades containing viruses isolated during the first wave (2014–2015) and the second wave (2016–2017) of the global dispersal of H5Nx were identified and labeled. Within the 2014–2015 sublineage, the viruses isolated in North Asia (Korea, Japan and Russia), Europe, North America and Taiwan were colored as blue, purple, green and red, respectively. (B) Time-scaled HA phylogeny of viruses responsible for 2014–2015 global outbreaks. Branch colors indicate inferred ancestral geographical regions of each branch. Sequences identified from wild birds were highlighted with dark dots on the tips. (C) Posterior probability distributions of the time of the most recent common ancestor (tMRCA) for viruses isolated in distinct countries during 2014–2015. Internal nodes corresponding to the position of tMRCAs were illustrated with dashed lines. (DOCX)

## Acknowledgments

The authors would like to express their deepest appreciation to the work team from Animal Health Inspection Division, BAPHIQ for the efforts in actively identifying outbreaks. We are also thankful of Dr. T.S. Chen, from the Endemic Species Research Institute (ESRI), COA, kindly providing the wetland database. The authors are also deeply appreciated to the work team of Remote Sensing, Division of Agricultural Chemistry, TARI for the efforts in identifying domestic waterfowl farms. The funding source of this study had no role in the study design, data collection, data analysis, data interpretation, or writing of the report. The corresponding author had full access to all the data in the study and had final responsibility for the decision to submit for publication.

## Author Contributions

**Conceptualization:** Day-Yu Chao.

**Data curation:** Wei-Shan Liang, Yu-Chen He, Tai-Hwa Shih, Horng-Yuh Guo.

**Formal analysis:** Wei-Shan Liang.

**Funding acquisition:** Day-Yu Chao.

**Investigation:** Yu-Chen He, Yao-Tsun Li, Gour-Shenq Kao, Horng-Yuh Guo, Day-Yu Chao.

**Methodology:** Hong-Dar Wu, Yao-Tsun Li, Day-Yu Chao.

**Project administration:** Tai-Hwa Shih, Gour-Shenq Kao, Horng-Yuh Guo, Day-Yu Chao.

**Resources:** Tai-Hwa Shih.

**Supervision:** Hong-Dar Wu, Gour-Shenq Kao, Day-Yu Chao.

**Validation:** Hong-Dar Wu, Day-Yu Chao.

**Writing – original draft:** Day-Yu Chao.

**Writing – review & editing:** Day-Yu Chao.

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
