## [Decision Letter · Decision Letter 0]

16 Apr 2020

PONE-D-20-01853

Ecological factors associated with persistent circulation of multiple highly pathogenic avian influenza viruses among poultry farms in Taiwan during 2015-17

PLOS ONE

Dear Dr. Chao,

Thank you for submitting your manuscript to PLOS ONE. After careful consideration, we feel that it has merit but does not fully meet PLOS ONE’s publication criteria as it currently stands. Therefore, we invite you to submit a revised version of the manuscript that addresses the points raised during the review process.

Substantial revisions should be made on the aspects that relate to the role of migratory birds. Indeed, many statements imply that migratory birds could have a role in the outbreaks describes in this study, but they all lack scientific support. Please consider removing the following statements:

- Line 41: "As an important stop-over site for migratory birds along the Asian flyway". It doesn't add anything to the sentence, only confusion regarding a potential role of migratory birds.

- Line 100: "on the Asian flyway of migrating birds,". Same remarks.

- Line 155: "when the migrating birds arrive in Taiwan". What species, what is the precise timing, where, etc. ?

- Line 170: "for the migratory season".

- Lines 406-408: "The HPAI-affected poultry farm outbreaks in Taiwan during 2015-2017 came in two waves: the first wave started in January, 2015 as migrating birds introduced HPAI H5N2, H5N8 and H5N3 to the island". More convincing evidence should be provided here (based on previously published studies or ecological and epidemiological data).

- Lines 410-412: "which was consistent with the arrival season of the migrating birds

(Supplementary Figure S3)."

- Lines 417-419: "Several studies have suggested that migratory birds played an important role in

long-distance transmission of HPAI H5Nx along the flyways(Global Consortium for H5N8 and Related Influenza Viruses, 2016, Bi et al., 2016)". There are also many studies that have shown that the "role of migratory birds" is a vague statement that requires more precision regarding species and timing of introduction of viruses via bird migration. Such a general statement is not acceptable.

- Lines 417-428 and Figure 4: I strongly disagree with this basic interpretation of the phylogenetic trees. The demonstration of virus spillover from wild to domestic birds requires further analyses than what is presented here. Again, information on migratory birds sampling (timing, location, birds species) should be provided, as well as contact/entry points between wild and domestic birds. More genetic data is also needed (full genome). This part of the study do not provide any information on the "ecological factors involved in persistent circulation of HPAI in poultry farms", which is the identified goal of the study. I strongly encourage to remove it.

Lines 595-610: This paragraph further highlight that the role of wild birds in the introduction and maintenance in Taïwan is unclear...

Also, figure quality should be improved. There inconsistencies in the reference format (numbers and then author names), in the main body of the manuscript.

We would appreciate receiving your revised manuscript by May 31 2020 11:59PM. To enhance the reproducibility of your results, we recommend that if applicable you deposit your laboratory protocols in protocols.io, where a protocol can be assigned its own identifier (DOI) such that it can be cited independently in the future. For instructions see: http://journals.plos.org/plosone/s/submission-guidelines#loc-laboratory-protocols

We look forward to receiving your revised manuscript.

Kind regards,

Camille Lebarbenchon

Academic Editor

PLOS ONE

Journal Requirements:

3. Please amend your authorship list in your manuscript file to include author "Gour-Shenq Kao".

4. We note that Figures 2 and 5 in your submission contain map images which may be copyrighted. All PLOS content is published under the Creative Commons Attribution License (CC BY 4.0), which means that the manuscript, images, and Supporting Information files will be freely available online, and any third party is permitted to access, download, copy, distribute, and use these materials in any way, even commercially, with proper attribution. For these reasons, we cannot publish previously copyrighted maps or satellite images created using proprietary data, such as Google software (Google Maps, Street View, and Earth). For more information, see our copyright guidelines: http://journals.plos.org/plosone/s/licenses-and-copyright.

a).    You may seek permission from the original copyright holder of Figure(s) [#] to publish the content specifically under the CC BY 4.0 license.

b).    If you are unable to obtain permission from the original copyright holder to publish these figures under the CC BY 4.0 license or if the copyright holder’s requirements are incompatible with the CC BY 4.0 license, please either i) remove the figure or ii) supply a replacement figure that complies with the CC BY 4.0 license. Please check copyright information on all replacement figures and update the figure caption with source information. If applicable, please specify in the figure caption text when a figure is similar but not identical to the original image and is therefore for illustrative purposes only.

Reviewers' comments:

Reviewer's Responses to Questions

**Comments to the Author**

1. Is the manuscript technically sound, and do the data support the conclusions?

Reviewer #1: Yes

Reviewer #2: Yes

2. Has the statistical analysis been performed appropriately and rigorously? 

Reviewer #1: Yes

Reviewer #2: I Don't Know

3. Have the authors made all data underlying the findings in their manuscript fully available?

Reviewer #1: Yes

Reviewer #2: No

4. Is the manuscript presented in an intelligible fashion and written in standard English?

Reviewer #1: Yes

Reviewer #2: Yes

5. Review Comments to the Author

Reviewer #1: Comments:

In this manuscript Liang et al. evaluated ecological and environmental factors that are associated with persistent circulation of highly pathogenic avian influenza A virus (HPAIV), especially H5N2 and H5N8. They employed satellite imaging technology for risk factors evaluation. They found four risk factors that were strongly associated with persistent circulation of H5N2 and H5N8 subtype viruses in Taiwan from 2015 to 2017. These four risk factors are high poultry farm density, poultry heterogeneity index, non-registered waterfowl flock density, and higher percentage of cropping land coverage.

Their analyses point to the importance of satellite remote sensing and clustering analysis for discovering environmental factors that may influence persistent circulation of H5Nx viruses in the poultry farms in Taiwan. The findings in this study are interesting and novel. However, there are still some issues and concerns that need to be properly addressed and clarified by the authors.

Major concerns

1. In this study the authors found four risk factors strongly associated with persistent circulation of H5N2 and H5N8 subtype viruses in Taiwan from 2015 to 2017. These four factors are high poultry farm density (allrD), poultry heterogeneity index (PHI), non-registered waterfowl flock density (nrwaterD) and higher percentage of cropping land coverage (allcrop).

In Table 3, the adjusted odds ratios (aOR) for the four risk factors vary from 17.46 (allrD) in 2015 to 1.04 (allcrop) in 2017, respectively. The authors should discuss, among these four risk factors, which factors are the most important in affecting persistent circulation of H5N2 and H5N8 viruses in Taiwan.

The four aORs in the year of 2015 are 6.80 (nrwaterD_high), 17.46 (allrD_high), 12.28 (PHI), and 1.36 (allcrop), respectively. And the four aORs in the year of 2017 are 9.17 (nrwaterD_high), 8.23 (allrD_high), 2.79 (PHI), and 1.04 (allcrop), respectively.

Among the four aORs in 2015, the highest aOR is 17.46 (allrD_high), whereas the highest aOR in 2017 is 9.17 (nrwaterD_high).

The authors should briefly discuss why the highest aOR changes substantially from 17.46 (allrD_high) in 2015 to 8.23 (allrD_high) in 2017, and whether the ranking of the aORs of the four risk factors are affected by the year of study. The authors should also explain why the aOR changed sharply from 12.28 (PHI) in 2015 to 2.79 (PHI) in 2017.

2. On lines 412 – 415, see “According to the subtypes of H5Nx identified from the poultry farms, H5N3 was only detected in 2015 and H5N6 only in 2017. However, H5N2 and H5N8 have been persistently detected among the poultry farms for three consecutive years since their first arrival in 2015 (Fig 3).”

The above statement is not accurate. The H5N8 virus could only be detected in geese in 2015 and before the summer of 2016, but not after the summer of 2016.

Specifically speaking, the H5N8 virus was not detected in geese in 2017. It is the H5N2 virus that persistently circulated in the chickens, ducks and geese in Taiwan in three years from 2015 to 2017. Based on the data presented in Figure 3, the H5N8 virus did not persistently circulate in geese after the summer of 2016, and was not detected in the whole year of 2017 as well. Therefore, the H5N8 virus was not detected in geese for the three consecutive years.

The authors should briefly discuss specifically why H5N3 was only detected in 2015 and H5N6 only in 2017, and should discuss potential environmental factors that might affect the spread of these two viruses, not just mention broadly “local environmental factors” or “geographical locations”.

3. In 2015 and 2016, the H5N8 was detected multiple times in chickens and ducks. However, in 2017 the H5N8 virus was only detected once or twice in chickens and ducks.

The authors should briefly explain why the H5N8 spread less efficiently in chickens and ducks in 2017 than its spread in 2016. What had changed for the four risk factors in 2017 compared to those in 2016?

Although the H5N8 was detected multiple times in geese in the year of 2015, it was not detected after the summer of 2016. What is more, the H5N8 virus was not detected in geese at all in the whole year of 2017.

The authors should discuss what environmental or ecological factors may play a role in preventing efficient spread of H5N8 virus in the geese after the summer of 2016 and in the year of 2017.

The authors claimed that among the chickens, duck and goose farms, the goose farms in 2015 are affected by HPAI H5Nx with the highest percentage (37.17%). However, the spread of H5N8 virus in geese was less efficiently than those in chickens and ducks after the summer of 2016 and in the year of 2017. The authors should give some explanations why that is the case.

Minor concerns

1. Whether LPAI H5N2 and HPAI H5N2 viruses co-circulated in the same hotspot poultry farms from 2015 to 2017?

If they did, whether co-circulation of LPAI H5N2 with the highly pathogenic H5Nx viruses may help maintain persistent circulation of HPAI H5N2 in the poultry farms?

2. The authors should discuss why the H5N2 virus circulated efficiently in chickens, ducks and geese from 2015 to 2017, whereas the H5N8 virus did less efficiently in geese after the summer of 2016 and in the whole year of 2017.

Reviewer #2: It would be better if the risk factors are indicated with odds rations and p values in abstract also.

Introduction part is a bit short and need to add some ecological parameters related to HPAI and elaborate information on the remote sensing.

On what basis the cut off for the farms size was made for the analysis and did you mentioned that in the paper?

Need to be careful in using the terms correlations and associations especially in line numbers 464 to 474..!

6. PLOS authors have the option to publish the peer review history of their article (what does this mean?). If published, this will include your full peer review and any attached files.

Reviewer #1: Yes: Yes

Reviewer #2: No

---

## [Author Response · Author response to Decision Letter 0]

28 May 2020

Dear Editors,

We are grateful for the critical comments provided by the reviewers to the manuscript. We address comments raised by the reviewers and list point by point answers to each comment as below. The revision at the manuscript has been labeled in red color to be easier to read. 

Comments from the Editor

Substantial revisions should be made on the aspects that relate to the role of migratory birds. Indeed, many statements imply that migratory birds could have a role in the outbreaks describes in this study, but they all lack scientific support. 

Reply:

Migratory birds are the natural reservoirs of avian influenza viruses. Between 2008 and 2010, highly pathogenic avian influenza (HPAI) H5 of the N1 subtype from the A/goose/Guangdong/1/96-like (Gs/GD) lineage generated novel reassortants by introducing other neuraminidase (NA) subtypes reported to cause most global outbreaks in poultry. Global Consortium for H5N8 and Related Influenza Viruses performed epidemiological investigations of waterfowl migration and poultry trade have shown that migratory birds can play a major role in the long-distance spread of avian influenza viruses resulting global poultry farm outbreaks as published in Science 2016 (reference #9). However, the reason why the transmission mechanism between migratory birds and domestic poultry is elusive mainly due to the low sampling rate of wild birds from the surveillance system implemented and different susceptibility of wild birds infected by different subtypes of HPAI. Monitoring HPAI H5N2 genetic changes based on active virological surveillance system in Taiwan has revealed that re-assortment of internal genes with multiple LPAI viruses from wild birds has occurred, although the HA and NA genes remain the same (BAPHIQ website ai.gov.tw). In particular, a new re-assorted HPAI H5N5 with HA gene belonging to clade 2.3.4.4 descendent with Taiwan local strains, was identified since September, 2019. Our virus isolation data further provided the direct scientific support of gene exchange between the domestic poultry and wild birds (recently being accepted to publish in the journal “Virus Evolution”). Furthermore, all four subtypes of H5Nx identified during 2015-2017 shared the same HA gene lineage derived from A/goose/Guangdong/1/96-like (Gs/GD) HPAI H5, which has never been identified before in Taiwan. All these evidences support the long-distance transmission of H5Nx HPAI into Taiwan by migratory birds. We are thankful for the editor’s comments and have revised the statements in the manuscript accordingly. 

Please consider removing the following statements:

- Line 41: "As an important stop-over site for migratory birds along the Asian flyway". It doesn't add anything to the sentence, only confusion regarding a potential role of migratory birds. 

Reply:

We have revised the sentence due to that we don’t have viral sequencing results from wild birds in 2015-2017. However, the HA gene of HPAI H5Nx from poultry farm suggested that they are highly clustered with A/goose/Guangdong/1/96-like (Gs/GD) lineage, which never been detected in Taiwan before, although the other routes of introduction cannot be excluded. The sentence is revised as such in the Abstract section, Line 40-44.

- Line 100: "on the Asian flyway of migrating birds,". Same remarks.

Reply:

The statement is deleted.

- Line 155: "when the migrating birds arrive in Taiwan". What species, what is the precise timing, where, etc. ?

Reply:

Migratory ducks come to Taiwan in September of each year including green-winged teal (Anas crecca), European wigeon (Anas penelope), and a few other Anas spp. such as the pintail (Anas acuta). Based on the wild bird AI surveillance (ref #13), the most common months of detecting avian influenza viruses are from September to April. We agree with the editor’s comment and the statement has been revised in Material and Methods section, Line 177.

- Line 170: "for the migratory season".

Reply:

We agree with the editor’s comment that the complete poultry farm census dataset isn’t for the migratory season although the satellite images were collected from August 2016 and April 2017 as stated in line 165. The statement is deleted. 

- Lines 406-408: "The HPAI-affected poultry farm outbreaks in Taiwan during 2015-2017 came in two waves: the first wave started in January, 2015 as migrating birds introduced HPAI H5N2, H5N8 and H5N3 to the island". More convincing evidence should be provided here (based on previously published studies or ecological and epidemiological data).

Reply:

Migratory birds are the natural reservoirs of avian influenza viruses. Between 2008 and 2010, highly pathogenic avian influenza (HPAI) H5 of the N1 subtype from the A/goose/Guangdong/1/96-like (Gs/GD) lineage generated novel reassortants by introducing other neuraminidase (NA) subtypes reported to cause most global outbreaks in poultry. Global Consortium for H5N8 and Related Influenza Viruses performed epidemiological investigations of waterfowl migration and poultry trade have shown that migratory birds can play a major role in the long-distance spread of avian influenza viruses resulting global poultry farm outbreaks as published in Science 2016 (reference #9). However, the reason why the transmission mechanism between migratory birds and domestic poultry is elusive mainly due to the low sampling rate of wild birds from the surveillance system implemented and different susceptibility of wild birds infected by different subtypes of HPAI. Recently, our virus isolation data (recently being accepted to publish in the journal “Virus Evolution”) provided the directly scientific support of gene exchange between the domestic poultry and wild birds. Furthermore, all four subtypes of H5Nx identified during 2015-2017 shared the same HA gene lineage derived from A/goose/Guangdong/1/96-like (Gs/GD) HPAI H5, which has never been identified before in Taiwan. All these evidences support the long-distance transmission of H5Nx HPAI into Taiwan by migratory birds.

We agree with the editor’s comment and the statement has been revised in Results section, Line 425-429 and added another paragraph in the discussion section, Line 600-632.

- Lines 410-412: "which was consistent with the arrival season of the migrating birds (Supplementary Figure S3)."

Reply: 

By following the editor’s comment, the statement has been deleted.

- Lines 417-419: "Several studies have suggested that migratory birds played an important role in long-distance transmission of HPAI H5Nx along the flyways(Global Consortium for H5N8 and Related Influenza Viruses, 2016, Bi et al., 2016)". There are also many studies that have shown that the "role of migratory birds" is a vague statement that requires more precision regarding species and timing of introduction of viruses via bird migration. Such a general statement is not acceptable.

Reply: 

Indeed, the reason why the transmission mechanism between migratory birds and domestic poultry is elusive mainly due to the low sampling rate of wild birds from the surveillance system implemented and different susceptibility of wild birds infected by different subtypes of HPAI. Precise information regarding species and timing of migratory birds relying on the following fragmented evidences. First, all four subtypes of H5Nx identified during 2015-2017 shared the same HA gene lineage derived from A/goose/Guangdong/1/96-like (Gs/GD) HPAI H5, which has never been identified before in Taiwan. Phylogenetic analyses, based on influenza HA genes, demonstrated that HPAI H5Nx viruses isolated in poultry farms during the first wave in Taiwan were closely related to two Japanese isolates identified from wild birds in late 2014 (Fig S6A and S6B). Along with other geographical areas involved in the same wave of global H5Nx expansion, HPAI viruses, discovered in Korea and Japan, could be the sources of viruses introduced to Taiwan in winter 2014 (Fig S6C). Secondly, Monitoring HPAI H5N2 genetic changes based on active virological surveillance system in Taiwan has revealed that re-assortment of internal genes with multiple LPAI viruses from wild birds has occurred, although the HA and NA genes remain the same (BAPHIQ website ai.gov.tw). In particular, a newly re-assorted HPAI H5N5 with HA gene belonging to clade 2.3.4.4 descendent with Taiwan local strains, was identified since September, 2019. Thirdly, our viral sequencing data from the field collections during 2019 provided the directly scientific support of gene exchange between the domestic poultry and wild birds (reference #53). Fourth, avian influenza virus monitoring in migratory birds has been performed in Taiwan since 1998 and was recently expanded to include the dead wild birds collected by the citizen scientists[13]. The results also showed the avian influenza viruses can be isolated from the domestic wild birds (https://ai.gov.tw/index.php?id=2017). 

We are thankful for the editor’s comment and the whole paragraph including the statement was moved to the discussion section, Line 600-632.

- Lines 417-428 and Figure 4: I strongly disagree with this basic interpretation of the phylogenetic trees. The demonstration of virus spillover from wild to domestic birds requires further analyses than what is presented here. Again, information on migratory birds sampling (timing, location, birds species) should be provided, as well as contact/entry points between wild and domestic birds. More genetic data is also needed (full genome). This part of the study do not provide any information on the "ecological factors involved in persistent circulation of HPAI in poultry farms", which is the identified goal of the study. I strongly encourage to remove it.

Reply: 

We are thankful for the editor’s comment and the whole paragraph including the statement was moved to the discussion section, Line 600-632.

Lines 595-610: This paragraph further highlight that the role of wild birds in the introduction and maintenance in Taiwan is unclear...

Reply: 

The role of wild birds in the introduction of HPAI in Taiwan could be different from the role of maintenance. The evidences suggested migratory birds play an important role in introducing novel HPAI subtypes into Taiwan rely on the following data. First, all four subtypes of H5Nx identified during 2015-2017 shared the same HA gene lineage derived from A/goose/Guangdong/1/96-like (Gs/GD) HPAI H5, which has never been identified before in Taiwan. Phylogenetic analyses, based on influenza HA genes, demonstrated that HPAI H5Nx viruses isolated in poultry farms during the first wave in Taiwan were closely related to two Japanese isolates identified from wild birds in late 2014 (Fig 6A and 6B). Along with other geographical areas involved in the same wave of global H5Nx expansion, HPAI viruses, discovered in Korea and Japan, could be the sources of viruses introduced to Taiwan in winter 2014 (Fig 6C). Secondly, Monitoring HPAI H5N2 genetic changes based on active virological surveillance system in Taiwan has revealed that re-assortment of internal genes with multiple LPAI viruses from wild birds has occurred, although the HA and NA genes remain the same (BAPHIQ website ai.gov.tw). In particular, a new re-assorted HPAI H5N5 with HA gene belonging to clade 2.3.4.4 descendent with Taiwan local strains, was identified since September, 2019. Thirdly, our viral sequencing data from the field collections during 2019 provided the directly scientific support of gene exchange between the domestic poultry and wild birds[50]. Fourth, avian influenza virus monitoring in migratory birds has been performed in Taiwan since 1998 and was recently expanded to include the dead wild birds collected by the citizen scientists[13]. The results also showed the avian influenza viruses can be isolated from the domestic wild birds[51] (https://ai.gov.tw/index.php?id=2017).

 On the contrary, the role of wild bird in the maintenance of HPAI rely on domestic ducks as natural reservoirs. The high poultry farm density and proximity of both domestic land and water poultry (high poultry heterogeneity index or PHI) provide a great opportunity for wild bird to transmit the virus from duck farms to chicken farms. 

We are thankful for the editor’s comment and this paragraph has been revised in the discussion section, Line 600-632.

Also, figure quality should be improved. There inconsistencies in the reference format

(numbers and then author names), in the main body of the manuscript.

Reply:

We are thankful for the editor’s comment and the figures with high resolution has been provided. Also, the references coherent with the journal’s format are revised. 

Comments from Reviewer #1

1. In this study the authors found four risk factors strongly associated with persistent circulation of H5N2 and H5N8 subtype viruses in Taiwan from 2015 to 2017. These four factors are high poultry farm density (allrD), poultry heterogeneity index (PHI), non-registered waterfowl flock density (nrwaterD) and higher percentage of cropping land coverage (allcrop). In Table 3, the adjusted odds ratios (aOR) for the four risk factors vary from 17.46 (allrD) in 2015 to 1.04 (allcrop) in 2017, respectively. The authors should discuss, among these four risk factors, which factors are the most important in affecting persistent circulation of H5N2 and H5N8 viruses in Taiwan. The four aORs in the year of 2015 are 6.80 (nrwaterD_high), 17.46 (allrD_high), 12.28 (PHI), and 1.36 (allcrop), respectively. And the four aORs in the year of 2017 are 9.17 (nrwaterD_high), 8.23 (allrD_high), 2.79 (PHI), and 1.04 (allcrop), respectively. Among the four aORs in 2015, the highest aOR is 17.46 (allrD_high), whereas the highest aOR in 2017 is 9.17 (nrwaterD_high). The authors should briefly discuss why the highest aOR changes substantially from 17.46 (allrD_high) in 2015 to 8.23 (allrD_high) in 2017, and whether the ranking of the aORs of the four risk factors are affected by the year of study. The authors should also explain why the

aOR changed sharply from 12.28 (PHI) in 2015 to 2.79 (PHI) in 2017.

Reply:

The main reason for the changes in aOR is due to the decline of the poultry farms with laboratory-confirmed H5Nx from 1,004 in 2015 to 37 in 2016 and 182 in 2017. However, it should be noted that although aOR seems different between years, the confidence intervals are overlapping. As for aOR of allrD_high is 17.46 in 2015 and 8.23 in 2017, the confidence interval is (95%CI: 5.91-74.86) for 2015 and (95%CI: 2.12-54.86). Also, as for aOR of PHI is 12.28 in 2015 and changed to 2.79 in 2017, the confidence interval is overlapping for both years with (95%CI: 5.02-31.14) for 2015 and (95%CI: 1.00-7.69). We are thankful the reviewer’s comment and the sentence has been revised in Discussion section, Line 575-576.

2. On lines 412 – 415, see “According to the subtypes of H5Nx identified from the poultry farms, H5N3 was only detected in 2015 and H5N6 only in 2017. However, H5N2 and H5N8 have been persistently detected among the poultry farms for three consecutive years since their first arrival in 2015 (Fig 3).” The above statement is not accurate. The H5N8 virus could only be detected in geese in 2015 and before the summer of 2016, but not after the summer of 2016. Specifically speaking, the H5N8 virus was not detected in geese in 2017. It is the H5N2 virus that persistently circulated in the chickens, ducks and geese in Taiwan in three years from 2015 to 2017. Based on the data presented in Figure 3, the H5N8 virus did not persistently circulate in geese after the summer of 2016, and was not detected in the whole year of 2017 as well. Therefore, the H5N8 virus was not detected in geese for the three consecutive years. The authors should briefly discuss specifically why H5N3 was only detected in 2015 and H5N6 only in 2017, and should discuss potential environmental factors that might affect the spread of these two viruses, not just mention broadly “local environmental factors” or “geographical locations”.

Reply:

Indeed, different subtypes of HPAI H5Nx with different intensity in different species were noticed (Fig 3). Specifically, H5N2 was persistently detected in the chickens, ducks and geese in Taiwan in three years from 2015 to 2017. However, H5N8 could only be detected in geese in 2015-2016, but not after the summer of 2016 and 2017. Ducks are inherently most resistant to diseases from HPAI viral infections than gallinaceous poultry, because they have retained the IFN-inducing RIG-I family of genes, lost from the jungle fowl that are the precursors of domestic chickens. Goose seems remain susceptible to HPAI infection although it retains the IFN-inducing RIG-I gene. The ducks usually don’t present any clinical symptoms after being infected by H5N2 or H5N8, compared to H5N3 and H5N6. The high domestic waterfowl to landfowl density ((high poultry heterogeneity index or PHI) could provide a good opportunity for the H5N8 virus to persistently circulate among waterfowl farms and spread to chicken farms when both types of farms are spatially close. We are thankful the reviewer’s comment and the sentence has been revised in Results section, Line 432-436, and Discussion section, Line 581-591. 

3. In 2015 and 2016, the H5N8 was detected multiple times in chickens and ducks. However, in 2017 the H5N8 virus was only detected once or twice in chickens and ducks. The authors should briefly explain why the H5N8 spread less efficiently in chickens and ducks in 2017 than its spread in 2016. What had changed for the four risk factors in 2017 compared to those in 2016? Although the H5N8 was detected multiple times in geese in the year of 2015, it was not detected after the summer of 2016. What is more, the H5N8 virus was not detected in geese at all in the whole year of 2017. The authors should discuss what environmental or ecological factors may play a role in preventing efficient spread of H5N8 virus in the geese after the summer of 2016 and in the year of 2017. The authors claimed that among the chickens, duck and goose farms, the goose farms in 2015 are affected by HPAI H5Nx with the highest percentage (37.17%). However, the spread of H5N8 virus in geese was less efficiently than those in chickens and ducks after the summer of 2016 and in the year of 2017. The authors should give some explanations why that is the case.

Reply:

Other than the virological and species susceptibility differences between H5N8 and H5N6 as stated in the previous comment, the factor why the H5N8 spread less efficiently in chickens and ducks in 2017 could be the control measures implemented by the government in Taiwan. After the first isolation of HPAI virus on January 5, 2015, the Central Emergency Operation Center (CEOC) was established and control strategies were immediately implemented. The control measures included four different components: (1) both passive and active surveillance and diagnosis, (2) culling and depopulation, (3) biosecurity and restriction on poultry movement, and (4) education[53,54]. Positive flocks or flocks within 3 km of an affected farm where the death rate was greater than 10% were destroyed without delay. Infected premises and equipment were cleaned and disinfected. Furthermore, movement of poultry and their products was restricted within a 1-3 km radius around an affected farm. With continuing reporting and confirmation of HPAI-affected farms, poultry slaughtering and marketing were suspended for four days starting on Jan 24, 2015. Marketing of poultry meat was suspended again from February 17-24, 2017, immediately after the confirmation of poultry affected by another novel HPAI subtype (H5N6). After necessary control measures were implemented, HPAI H5N3 was officially eliminated from Taiwan on May 22, 2015 and H5N6 was eliminated on May 5, 2017. However, due to the continuous circulation of HPAI H5N2 and H5N8, the active surveillance was not only applied to the surrounding poultry farms, but also included the rendering factories and slaughter houses for unusual numbers of dead poultry to ensure disease detection. The certificate of negative results based on the laboratory examination of poultry before shipped to the slaughter houses was mandated for the counties with the highest numbers of outbreak farms during Nov.-Mar. when the migrating birds arrive in Taiwan. Finally, it took three years for H5N8 to control and was officially eliminated after Feb. 22, 2018.

We are thankful the reviewer’s comment and the paragraph providing the explanation has been revised in the Discussion section, Line 647-669.

4. Whether LPAI H5N2 and HPAI H5N2 viruses co-circulated in the same hotspot poultry farms from 2015 to 2017? If they did, whether co-circulation of LPAI H5N2 with the highly pathogenic H5Nx viruses may help maintain persistent circulation of HPAI H5N2 in the poultry farms?

Reply:

Based on the surveillance data, LPAI H5N2 and HPAI H5N2 viruses did co-circulate in the same hotspot poultry farms from 2015 to 2017 and it may help to maintain the persistent circulation of HPAI H5N2 in the poultry farms. We are thankful the reviewer’s comment and the paragraph providing the explanation has been revised in the Discussion section, Line 669-686.

5. The authors should discuss why the H5N2 virus circulated efficiently in chickens, ducks and geese from 2015 to 2017, whereas the H5N8 virus did less efficiently in geese after the summer of 2016 and in the whole year of 2017.

Reply:

The reasons that HPAI H5N2 continue to circulate and couldn’t be eliminated so far in Taiwan are probably due to at least two factors. First, domestic ducks with low pathogenicity to HPAI play a role in preserving the viruses. High HPAI H5 seropositivity in duck farms, based on the results of avian influenza sero-surveillance system implemented by BAPHIQ, confirmed that domestic ducks play an important role as the reservoir for sustained transmission and influenza virus re-assortment events among poultry farms. Waterfowl play an important role as an influenza virus reservoir, harboring genes that might contribute to HPAIVs. These viruses have the potential to be introduced into landfowl holdings and cause severe disease in gallinaceous poultry through direct contact between poultry and infected wild birds or through indirect contact with fomites (e.g., water, feed bedding material, boots, and wheels of vehicles) contaminated with wild bird feces [9,49]. Secondly, the co-circulation of LPAI and HPAI H5N2 at the same time among the poultry farms. After four years of control efforts, this new subtype of HPAI H5N2 was shown to be particularly difficult to eliminate and continues to persist, together with LPAI Mexican-lineage H5N2, in domestic poultry farms. Taiwan has implemented intense surveillance and control measures since 2015 and has reported more poultry outbreak events to OIE than any other country (Fig 6).

We are thankful the reviewer’s comment and the paragraph providing the explanation has been revised in the Discussion section, Line 669-686.

Comments from Reviewer #2

1. It would be better if the risk factors are indicated with odds rations and p values in abstract also.

Reply:

We are thankful for the reviewer’s comments and the odds rations and p values were added in the abstract section, Line 51-57. 

2. Introduction part is a bit short and need to add some ecological parameters related to HPAI and elaborate information on the remote sensing.

Reply:

We are thankful for the reviewer’s comments and additional information on remote sensing and ecological parameters related to this study have been added to the Introduction section, Line 98-114.

3. On what basis the cut off for the farms size was made for the analysis and did you mentioned that in the paper?

Reply:

We didn’t set the cutoff for the farm size but the farm density was trisected based on generalized additive model (GAM) analysis as stated in the Material and Methods section, Line 336-341.

4. Need to be careful in using the terms correlations and associations especially in line numbers 464 to 474..!

Reply:

Thanks for the reviewer’s reminding here. Indeed, association is a relationship between two random variables which refers to rather a general relationship without quantitative measurements or a causal relationship. On the other hand, correlation is a measure of association and statistical models are used to provide the measurement of the magnitude of the correlation. 

We agree with the reviewer’s comment and the statements in Line 363 to 474 are more appropriate to use “correlation”. No further revision was made here.

---

## [Decision Letter · Decision Letter 1]

10 Jul 2020

Ecological factors associated with persistent circulation of multiple highly pathogenic avian influenza viruses among poultry farms in Taiwan during 2015-17

PONE-D-20-01853R1

Dear Dr. Chao,

We’re pleased to inform you that your manuscript has been judged scientifically suitable for publication and will be formally accepted for publication once it meets all outstanding technical requirements.

Kind regards,

Camille Lebarbenchon

Academic Editor

PLOS ONE

Additional Editor Comments (optional):

Reviewers' comments:

Reviewer's Responses to Questions

**Comments to the Author**

1. If the authors have adequately addressed your comments raised in a previous round of review and you feel that this manuscript is now acceptable for publication, you may indicate that here to bypass the “Comments to the Author” section, enter your conflict of interest statement in the “Confidential to Editor” section, and submit your "Accept" recommendation.

Reviewer #1: All comments have been addressed

Reviewer #2: All comments have been addressed

2. Is the manuscript technically sound, and do the data support the conclusions?

Reviewer #1: Yes

Reviewer #2: Yes

3. Has the statistical analysis been performed appropriately and rigorously? 

Reviewer #1: No

Reviewer #2: Yes

4. Have the authors made all data underlying the findings in their manuscript fully available?

Reviewer #1: Yes

Reviewer #2: Yes

5. Is the manuscript presented in an intelligible fashion and written in standard English?

Reviewer #1: Yes

Reviewer #2: Yes

6. Review Comments to the Author

Reviewer #1: The authors have carefully addressed the issues and concerns raised by the other reviewer and myself. I do not have more comments about the manuscript.

Reviewer #2: The manuscript has been improved from the previous version. The authors had made some changes to the statistical analysis. For the data, the authors had worked on it to make it better.

7. PLOS authors have the option to publish the peer review history of their article (what does this mean?). If published, this will include your full peer review and any attached files.

Reviewer #1: No

Reviewer #2: **Yes: **Tulsi Ram Gompo

---

## [Editor Report · Acceptance letter]

15 Jul 2020

PONE-D-20-01853R1 

Ecological factors associated with persistent circulation of multiple highly pathogenic avian influenza viruses among poultry farms in Taiwan during 2015-17 

Dear Dr. Chao:

I'm pleased to inform you that your manuscript has been deemed suitable for publication in PLOS ONE. Congratulations! Your manuscript is now with our production department. 

Kind regards, 

on behalf of

Dr. Camille Lebarbenchon 

Academic Editor

PLOS ONE